# LEARNING GUARANTEES FOR GRAPH CONVOLUTIONAL NETWORKS ON THE STOCHASTIC BLOCK MODEL

**Wei Lu**
Department of Mathematics
Brandeis University
Waltham, MA 02453, USA
`luwei@brandeis.edu`

## ABSTRACT

An abundance of neural network models and algorithms for diverse tasks on graphs have been developed in the past five years. However, very few provable guarantees have been available for the performance of graph neural network models. This state of affairs is in contrast with the steady progress on the theoretical underpinnings of traditional dense and convolutional neural networks. In this paper we present the first provable guarantees for one of the best-studied families of graph neural network models, Graph Convolutional Networks (GCNs), for semi-supervised community detection tasks. We show that with high probability over the initialization and training data, a GCN will efficiently learn to detect communities on graphs drawn from a stochastic block model. Our proof relies on a fine-grained analysis of the training dynamics in order to overcome the complexity of a non-convex optimization landscape with many poorly-performing local minima.

## 1 INTRODUCTION

There is presently a large gap between what can be accomplished in practice using deep learning, and what can be satisfactorily explained and predicted by the theory of deep learning. Nevertheless, the past several years have seen substantial developments in the theory of deep learning (Ge et al., 2017; Brutzkus & Globerson, 2017; Zhang et al., 2019a; Goel et al., 2020; Chen et al., 2020a).

One factor contributing to the gap between the theory and practice of traditional NNs is that real-world data sets tend to have complex structure that is difficult to capture with formal definitions. For example, popular image classification models are capable of memorizing arbitrary data (Zhang et al., 2016), and yet they exhibit astonishing generalization performance on accurately-labeled natural images. Hence, any rigorous proof of the observed generalization performance of deep learning models on image classification tasks will necessarily require assumptions about the data that are sharp enough to separate random inputs from natural images. Because of the difficulty of giving an adequate characterization of real-world data, much of the recent progress in deep learning theory has instead focused on proving results using very simple (e.g. Gaussian) input distributions or in distribution-free settings (Ge et al., 2017; Brutzkus & Globerson, 2017; Zhang et al., 2019a; Vempala & Wilmes, 2019).

Compared to traditional feed-forward (dense, convolutional, etc.) NNs, the theory of *graph* neural networks (GNNs) is still in its infancy. On the other hand, it appears substantially easier to give plausible descriptions of the combinatorial structure of real-world graph data sets than, e.g., to characterize the distribution of natural images (Drobyshevskiy & Turdakov, 2019). We therefore believe that GNNs offer a natural setting for developing provable guarantees that are able to capture the power of deep learning on real-world datasets. In this paper, we contribute to that goal by giving the first rigorous guarantees of efficient semi-supervised learning of stochastic block models via a GNN.

## 1.1 Graph neural networks

Many natural datasets for diverse machine learning problems have a graph structure, including social networks, molecular structures, and transit networks. In order to efficiently exploit such combinatorial structure, a variety of GNN models have been proposed, tuned for different kinds of tasks. A number of taxonomies of GNN models have been proposed (Zhou et al., 2018; Wu et al., 2021); one of the most essential differences between different GNN models is whether they are meant to label the graph as a whole, or to label individual components of the graph, particularly vertices.

From a theoretical perspective, the best understood tasks for GNNs concern labeling the graph as a whole, for example for the task of classifying a graph by its isomorphism type (Sato, 2020). In particular, it has been established that many GNN models are of comparable power to various versions of the Weisfeiler-Leman hierarchy[1] (Xu et al., 2018; Morris et al., 2019).

Some progress has also been made on the theory of GNNs for vertex-labeling tasks. Recent works by Sato et al. describe the representational power of certain GNN models for tasks such as computing minimum vertex covers (Sato et al., 2019). Garg et al. also give bounds on the representational power of GNN models, as well as using Rademacher bounds to estimate the generalization ability of GNNs (Garg et al., 2020).

Our results concern the task of semi-supervised community detection. In this problem, each vertex belongs to one community, and some subset of the vertices are labeled according to their community membership. The task is to classify the community membership of the remaining vertices. This task has been one of the most intensively studied problems in the GNN literature, but there have not yet been any provable guarantees on the performance of proposed models.

We study (spatial-based) graph convolutional models similar to the GCN model proposed in Kipf & Welling (2017). A single layer of such a model computes weights at each node by aggregating the weights at neighboring nodes and applying an activation function with learned parameters, e.g., a linear map followed by a ReLU. Many variations on this theme, including various sophisticated training regimes, have been proposed (Chen et al., 2017; Gao et al., 2018; Li et al., 2018; Zhang et al., 2019b; Chen et al., 2018), but no provable guarantees have been available for the performance of such models on natural data distributions, until the present work.

## 2 Main Results

One motivation for GNNs as a target for progress in deep learning theory is that there are well-studied graph distributions that plausibly capture some of the structure of real-world data (Drobyshevskiy & Turdakov, 2019). For example, even fairly simple preferential attachment models plausibly capture some of the essential structure of the web (Kumar et al., 2000). Other graph models naturally capture community structures, the simplest of which is the Stochastic Block Model (SBM) (Holland et al., 1983). A graph is sampled from a SBM by first partitioning vertices into communities (with fixed or random sizes). Two vertices are connected with probability $p$ if they belong to the same community and probability $q$ if they belong to different communities.

In this paper, we consider the case of an SBM with two equal-sized communities in which vertices have label 0 and 1 respectively. We denote the label of vertex $x$ by $\ell(x) \in \{0, 1\}$. The graphs are parameterized as $\text{SBM}(n, p, q)$ where $n$ is the number of vertices, $p$ is the probability of an intra-community connection, and $q$ is the probability of a cross-community connection. We allow $n$ to vary (but will require it to be sufficiently large), while $p$ and $q$ are of the form $p = \frac{a \log^3 n}{n}$ and $q = \frac{b \log^3 n}{n}$ for some fixed constants $a > b$. In the semi-supervised setting, the community labels of some portion of the labels are revealed. We assume the label of each vertex is revealed independently with probability $\lambda$. The input-layer features at a vertex $x$ is $(0, 0)$ if its label is not revealed, $(1, 0)$ if its label is revealed to be 0, and $(0, 1)$ if its label is revealed to be 1.

**Assumption 2.1** (Sparse Stochastic Block Model)**.** *The probabilities of intra and cross-community connections are $p = \frac{a \log^3 n}{n}$ and $q = \frac{b \log^3 n}{n}$, where $a > b$ are constants.*

---

[1]Weisfeiler-Leman hierarchy is a polynomial-time iterative algorithm which provides a necessary but insufficient condition for graph isomorphism.

We study the problem of recovering the communities from such graphs using GNN models. Of course, recovering the communities of an SBM graph has been well-studied and its computational complexity is fully understood in most cases (Abbe & Sandon, 2015; Kawamoto et al., 2019). SBM models are therefore a natural test-case for understanding the power of GNN models for learning community structure, and experimental studies have been done in this setting (Chen et al., 2020b; Yadav et al., 2019). (Abbe et al., 2014) shows a sharp threshold in the task of community recovery: $(\sqrt{p} - \sqrt{q})\sqrt{\frac{n}{\log n}} > \sqrt{2}$. This threshold clearly holds for our case (at sufficiently large values of $n$), since $p = \frac{a \log^3 n}{n}, q = \frac{b \log^3 n}{n}$ and $a > b$. The contribution here is not to learn the community models. Rather it's showing that (multi-layer) GCNs solve the classification problem, which is very much not trivial (it is non-convex, and the training loss curve is empirically non-monotonic).

Our GNN models will be trained on a graph or several graphs generated by the $SBM(n, p, q)$ model, and seek to understand their accuracy on arbitrary $SBM(n, p, q)$ graphs not necessarily in the training set but with the same parameters $a, b$ determining $p$ and $q$ (with $n$ allowed to vary).

In particular, we study spatial-based graph convolutional models along the lines of the Graph Convolutional Networks (GCN) introduced in (Kipf & Welling, 2017). Each layer of the model computes a feature vector at every vertex of an input graph based on features of nearby vertices in the previous layer. A typical layer-wise update rule is of the form
$$X^{(k+1)} = \phi\big(\hat{A}X^{(k)}W^{(k)}\big),$$
where

- $\hat{A}$ is a suitably-normalized adjacency matrix of shape $n \times n$ where $n$ is the number of vertices. Usually $\hat{A}$ includes self-loops.
- $X^{(k)}$ gives the feature vector in the $k$-th layer at each vertex as a matrix of shape $n \times m_k$, where $m_k$ is the number of features in layer $k$.
- $\phi$ is an activation function, such as the ReLU.
- $W^{(k)}$ are the trainable weights in the $k$-th layer, a matrix of shape $m_k \times m_{k+1}$.

In our version of this model, we define $\hat{A} = \frac{1}{\frac{n}{2}(p+q)}\tilde{A}$, where $\tilde{A} = A + I$, $A$ is the adjacency matrix of a given graph, and $I$ is the identity matrix. For the given $SBM(n, p, q)$, a randomly selected vertex has $\frac{n}{2}(p + q)$ neighbors in expectation, so $\hat{A}$ is obtained by normalizing each row of $A + I$ with the average size of a neighborhood. Since very deep GCN models seem to provide little empirical benefit (Li et al., 2018), we use a single hidden layer with a softmax output layer. Furthermore, we introduce a bias term $B$ at the second layer. So the model has the following form:
$$f(X, A) = \text{softmax}\big(\hat{A}\phi\big(\hat{A}XW^{(0)}\big)W^{(1)} + B\big)$$
$$= \text{softmax}\left(\frac{4}{n^2(p+q)^2}\tilde{A}\phi\big(\tilde{A}XW^{(0)}\big)W^{(1)} + B\right), \tag{1}$$

where $X$ is the input feature of the graph and $W^{(0)}, W^{(1)}$ and $B$ are trainable parameters. Let $h$ denote the number of hidden features, which equals the number of columns of $W^{(0)}$ and the number of rows of $W^{(1)}$.

We define the accuracy of the model as the probability of predicting correctly the label of a single vertex in a randomly generated $SBM(n, p, q)$ graph where the label of each vertex is revealed with probability $\lambda$. We can now state our main result.

**Theorem 2.2.** *For any $\epsilon > 0$ and $\delta > 0$, given a GCN model with $\frac{1}{\delta} \leq h \leq n$ hidden features and with parameters initialized independently from $N(0, 1)$, if training graphs are sampled from $SBM(n, p, q)$ with $n \geq \max(\Omega(\frac{1}{\epsilon})^2, \Omega(\frac{1}{\delta}))$ and the label of each vertex revealed with probability $\lambda$, and if the model is trained by coordinate descent for $k = O(\log\log \frac{1}{\epsilon})$ epochs, then with probability $\geq 1 - \delta$, the model achieves accuracy $\geq 1 - 4\epsilon$.*

*Remark.* We treat $\lambda$ as constants, so it is omitted in the big O and $\Omega$ notation in the sampling and training complexity.

We emphasize that the novelty of this theorem is not in learning two-class SBM models as such; this is a long-solved problem. Instead, this is the first proof of efficient learning for a GCN on semi-supervised community detection tasks using a natural family of random graph models.

## 3 PRELIMINARIES

In this section, we first introduce notations (a table of notations is also shown in the appendix for readers' convenience) and some interpretations. Then we introduce the structure of the paper. Given a vertex $y$, denote the row of $\tilde{A}X$ corresponding to $y$ as $(t_0^y, t_1^y)$, so $t_0^y$ and $t_1^y$ give the numbers of neighbors of $y$ (including perhaps $y$ itself) with revealed labels in class 0 and class 1 respectively. Let

$$
W^{(0)} = \begin{pmatrix} \alpha_1 & \alpha_2 & \cdots & \alpha_h \\ \alpha_1' & \alpha_2' & \cdots & \alpha_h' \end{pmatrix}, \; W^{(1)} = \begin{pmatrix} \beta_1 & \beta_1' \\ \beta_2 & \beta_2' \\ \vdots & \vdots \\ \beta_h & \beta_h' \end{pmatrix}, \; B = \begin{pmatrix} b_0 & b_1 \\ b_0 & b_1 \\ \vdots & \vdots \\ b_0 & b_1 \end{pmatrix}.
$$

Then $\alpha_i t_0^y + \alpha_i' t_1^y, 1 \leq i \leq h$ gives $h$ features of vertex $y$ in the hidden layer. The inner product of the $y$-th row of $\phi(\tilde{A}XW^{(0)})$ and the columns of $W^{(1)}$ gives weighted sums of features of $y : \sum_{i=1}^h \beta_i \phi(\alpha_i t_0^y + \alpha_i' t_1^y)$ and $\sum_{i=1}^h \beta_i' \phi(\alpha_i t_0^y + \alpha_i' t_1^y)$, where $\phi$ represents the ReLU function.

Given a vertex $x$, the row of $\hat{A}\phi(\hat{A}XW^{(0)})W^{(1)}$ corresponding to $x$ is denoted by $(f_0(x), f_1(x))$ and is of the form

$$
\left( \frac{4}{n^2(p+q)^2} \sum_{y \in G} \mathbb{1}[y \sim x] \sum_{i=1}^h \beta_i \phi(\alpha_i t_0^y + \alpha_i' t_1^y), \frac{4}{n^2(p+q)^2} \sum_{y \in G} \mathbb{1}[y \sim x] \sum_{i=1}^h \beta_i' \phi(\alpha_i t_0^y + \alpha_i' t_1^y) \right),
$$

(2)

where $\mathbb{1}[y \sim x]$ is equal to 1 if $y$ and $x$ are connected, 0 otherwise. Denote

$$
f_0^i(x) := \frac{4\beta_i}{n^2(p+q)^2} \sum_{y \in G} \mathbb{1}[y \sim x]\phi(\alpha_i t_0^y + \alpha_i' t_1^y) \quad f_1^i(x) := \frac{4\beta_i'}{n^2(p+q)^2} \sum_{y \in G} \mathbb{1}[y \sim x]\phi(\alpha_i t_0^y + \alpha_i' t_1^y),
$$

so $f_0(x) = \sum_{i=1}^h f_0^i(x)$ and $f_1(x) = \sum_{i=1}^h f_1^i(x)$.

Denote $g_j(x) := f_j(x) + b_j, j = 0, 1$, where $(g_0(x), g_1(x))$ represents the logit of the model corresponding to $x$. Denote $\Delta(x) := g_0(x) - g_1(x)$. In order to make correct predictions, we need $\Delta(x) > 0$ when $\ell(x) = 0$ and $\Delta(x) < 0$ when $\ell(x) = 1$.

The bias term $B$ is useful in our analysis because its derivative controls how imbalanced the current loss is between the classes. In training we consider the cross-entropy loss denoted as $L$, and have

$$
\mathbb{E}[\frac{\partial L}{\partial b_0}] = -\mathbb{E}[\frac{\partial L}{\partial b_1}] = -\frac{1}{2}(\mathbb{E}[Z|\ell(x) = 0] - \mathbb{E}[Z|\ell(x) = 1]),
$$

where $Z = \frac{\exp(g_{1-\ell(x)}(x))}{\exp(g_0(x)) + \exp(g_1(x))}$. $Z$ can be regarded as a measure of wrong prediction: the numerator is the exponential of the output corresponding to the wrong label and the denominator is a normalizer. It is easy to see that $Z > \frac{1}{2}$ if the prediction is wrong; $Z < \frac{1}{2}$ if prediction is correct. When $\left|\mathbb{E}[\frac{\partial L}{\partial b_0}]\right| \approx 0$, the model's loss is balanced in the sense of that $\left|\mathbb{E}[Z|\ell(x) = 0] - \mathbb{E}[Z|\ell(x) = 1]\right| \approx 0$. In order to have balanced performance in every epoch, we train the model through coordinate descent instead of conventional gradient descent. Specifically, in each epoch we first update $b_0$ and $b_1$ until $\left|\mathbb{E}[\frac{\partial L}{\partial b_0}]\right|$ is smaller than some threshold. Then we update the other parameters.

In order to make a learning guarantee of the model, we need a high probability estimation of $\Delta(x)$. In Section 4, we show that $\Delta(x)$ is concentrated at one of two values, denoted by $\mu_0$ and $\mu_1$, for $\ell(x) = 0$ and 1 respectively. The proof depends on different parameter regimes of hidden neurons. Furthermore, to avoid the overlap between the concentration range of $\Delta(x)$, we also show the separation between $\mu_0$ and $\mu_1$. In Section 5, we analyze the dynamics of hidden neurons throughout training to show that the concentration and separation improve at a controlled rate. Based on this information, in Section 6 we prove the main theorem. Section 7 shows some experimental results to verify our theory. The paper ends with future directions in Section 8.

## 4 CONCENTRATION AND SEPARATION OF OUTPUT

In this section we show that $\Delta(x)$ is concentrated at $\mu_0$ and $\mu_1$ and their separation. The difference of the logits is

$$\Delta(x) = g_0(x) - g_1(x) = f_0(x) - f_1(x) + b_0 - b_1 = \sum_{i=1}^{h} \Delta_i(x) + b_0 - b_1,$$

where

$$\Delta_i(x) = f_0^i(x) - f_1^i(x) = \frac{4(\beta_i - \beta_i')}{n^2(p+q)^2} \sum_{y \in G} \mathbb{1}[y \sim x] \phi(\alpha_i t_0^y + \alpha_i' t_1^y).$$

For brevity, we write $\Delta(x)$ as $\Delta$ and $\Delta_i(x)$ as $\Delta_i$. In order to estimate $\Delta$, we need to estimate each $\Delta_i, 1 \leq i \leq h$.

We denote the high probability estimate of $\Delta$ as $\mu_0$ and $\mu_1$ for $\ell(x) = 0$ and 1 respectively. Our fine-grained analysis of the dynamics of coordinate descent on GCNs relies on a classification of neurons into three families based on the sign and scale of the parameters: "good type", "bad type" and "harmless type". The names also indicate whether the neuron has positive contribution to the value of $\mu_0 - \mu_1$. We show that "good type" neuron makes positive contribution; the contribution of "bad type" neuron is negative but lower bounded; "harmless type" neuron's contribution is non negative (see Corollary A.4 and the remark following it). We will specifically describe parameter regime of each type in the following subsections. We analyze the dynamics of these types throughout coordinate descent in the next section. First we give some definitions.

**Definition 1.** For $1 \leq i \leq h$, we call $(\alpha_i, \alpha_i', \beta_i, \beta_i')$ the $i$-**th neuron** of the model, where $(\alpha_i, \alpha_i')^\top$ is the $i$-th column of $W^{(0)}$, $(\beta_i, \beta_i')$ is the $i$-th row of $W^{(1)}$.

**Definition 2.** We say that the $i$-th neuron is **order-aligned** if $(\alpha_i - \alpha_i')(\beta_i - \beta_i') > 0$, otherwise we say it is **order-misaligned**.

### 4.1 CLASSIFICATION OF NEURON PARAMETER REGIMES

We say the $i$-th neuron is of **"good type"** if it satisfies either $(G_1)$ or $(G_2)$ below. (There is also the symmetric case obtained by switching $\alpha_i$ with $\alpha_i'$ and $\beta_i$ with $\beta_i'$. For brevity, we only consider the cases that $\alpha_i > \alpha_i'$. This applies to the "bad" and "harmless" types below as well). Neurons in this type are order-aligned and both $\alpha_i$ and $\alpha_i'$ are positive or the ratio between $\alpha_i$ and $\alpha_i'$ is large enough.

$$\alpha_i > \alpha_i' > 0 \text{ and } \beta_i > \beta_i' \tag{$G_1$}$$

$$\alpha_i > 0 > \alpha_i', \left|\frac{\alpha_i}{\alpha_i'}\right| > 1 \text{ and } \beta_i > \beta_i' \tag{$G_2$}$$

We say the $i$-th neuron is of **"bad type"** if it satisfies either $(B_1)$, $(B_2)$ or $(B_3)$. Neurons in this type are order-misaligned and $\alpha_i, \alpha_i'$ are either both positive or have the opposite signs.

$$\alpha_i > \alpha_i' > 0 \text{ and } \beta_i < \beta_i' \tag{$B_1$}$$

$$\alpha_i > 0 > \alpha_i', \left|\frac{\alpha_i}{\alpha_i'}\right| > \frac{q}{p}(1 + \log^{-\frac{1}{3}} n) \text{ and } \beta_i < \beta_i' \tag{$B_2$}$$

$$\alpha_i > 0 > \alpha_i', \left|\frac{\alpha_i}{\alpha_i'}\right| \leq \frac{q}{p}(1 + \log^{-\frac{1}{3}} n) \tag{$B_3$}$$

We say that the $i$-th neuron is of **"harmless type"** if it satisfies either $(H_1)$ or $(H_2)$:

$$\alpha_i > 0 > \alpha_i', \left|\frac{\alpha_i}{\alpha_i'}\right| \in (\frac{q}{p}(1 + \log^{-\frac{1}{3}} n), 1] \text{ and } \beta_i > \beta_i' \tag{$H_1$}$$

$$\alpha_i \leq 0 \text{ and } \alpha_i' \leq 0 \tag{$H_2$}$$

### 4.2 CONCENTRATION AND SEPARATION

**Theorem 4.1.** *If the $i$-th neuron is of "good type" satisfying $(G_1)$ or of "bad type" satisfying $(B_1)$, then for $\ell(x) = 0$:*

$$\mathbb{P}[\left|\Delta_i - \frac{\lambda(\beta_i - \beta_i')}{(p+q)^2}[(p^2 + q^2)\alpha_i + 2pq\alpha_i']\right| \leq$$

$$(\alpha_i - \alpha_i')(\beta_i - \beta_i')O(\log^{-\frac{1}{2}} n)|\ell(x) = 0] \geq 1 - O\left(\frac{1}{n^2}\right),$$

*for $\ell(x) = 1$:*

$$\mathbb{P}[|\Delta_i - \frac{\lambda(\beta_i - \beta_i')}{(p+q)^2}[2pq\alpha_i + (p^2 + q^2)\alpha_i']| \leq$$

$$(\alpha_i - \alpha_i')(\beta_i - \beta_i')O(\log^{-\frac{1}{2}} n)|\ell(x) = 1] \geq 1 - O(\frac{1}{n^2}).$$

*Similar concentration hold for neurons satisfying $(G_2)$, $(B_2)$ and $(B_3)$, and for neurons of "harmless type."*

We apply the method of bounded differences to show the concentration. The details are shown in the appendix.

Given the concentration of $\Delta_i$ for each type of neurons, we estimate the concentration of the output $\Delta = \sum_{i=1}^{h} \Delta_i + b_0 - b_1$. For the $i$-th neuron, we denote the high-probability estimate of $\Delta_i$ given in the statement of Theorem 4.1 as $m_0^i$ when $\ell(x) = 0$ and $m_1^i$ when $\ell(x) = 1$. By union bound, we have the following corollary.

**Corollary 4.2.** *Given a vertex $x \in G$ with label unrevealed, we have*

$$\mathbb{P}[|\Delta - \mu_j| \leq \delta | \ell(x) = j] \geq 1 - O(\frac{1}{n}), \tag{3}$$

*where*

$$\mu_j = (\sum_{i=1}^{h} m_j^i) + b_0 - b_1, \ j = 0, 1 \qquad \delta = \sum_{i=1}^{h} |\alpha_i - \alpha_i'||\beta_i - \beta_i'|O(\log^{-\frac{1}{2}} n).$$

For any $\epsilon > 0$, we require the probability of concentration in (3) to be at least $1 - \tilde{\epsilon}$, where $\tilde{\epsilon} = o(\epsilon)$. If we choose $\tilde{\epsilon} = \epsilon^2$, then we set $1 - O(\frac{1}{n}) \geq 1 - \epsilon^2$, i.e. $n \geq \Omega(\frac{1}{\epsilon})^2$. Our following analysis will be based on this condition.

From Theorem 4.1, we have the following result about the value of $m_0^i - m_1^i$.

**Corollary 4.3.**     • *If the $i$-th neuron is of "good type" and satisfies $(G_1)$, then*

$$m_0^i - m_1^i = \lambda|\alpha_i - \alpha_i'||\beta_i - \beta_i'|\left(\frac{p-q}{p+q}\right)^2.$$

• *If the $i$-th neuron is of "bad type" and satisfies $(B_1)$, then*

$$m_0^i - m_1^i = -\lambda|\alpha_i - \alpha_i'||\beta_i - \beta_i'|\left(\frac{p-q}{p+q}\right)^2.$$

• *If the $i$-th neuron is of "harmless type" and satisfies $(H_1)$, then*

$$m_0^i - m_1^i = \lambda|\beta_i - \beta_i'||p\alpha_i + q\alpha_i'|\frac{p-q}{(p+q)^2}.$$

Similar results for neurons satisfying $(G_2), (B_2), (B_3)$ and $(H_1)$ are stated in the appendix, along with the proof.

*Remark.*     • As we can see from Corollary 4.3, the value of $m_0^i - m_1^i$ is positive for "good type" neurons, non-negative for "harmless type" neurons and may be negative (but lower bounded) for "bad type" neurons. Since positive values of $m_0^i - m_1^i$ decrease the loss of the model, this explains the names for the types of neurons.

• $m_0^i - m_1^i$ is proportional to $|\alpha_i - \alpha_i'||\beta_i - \beta_i'|$. In the next section, we analyze the dynamics of the parameters $\alpha_i, \alpha_i', \beta_i, \beta_i'$. Using our understanding of these dynamics, in Theorem 6.2 we present a refined result about the separation of output which only depends on the initialization of parameters.

• Let $c := \mu_0 - \mu_1 = \sum_{i=1}^{h}(m_0^i - m_1^i)$. By the two corollaries above, we have $\delta = o(|c|)$. The balanced loss guaranteed by the bias term and the coordinate descent scheme ensure that $\mu_0 = \Omega(c)$ and $\mu_1 = \Omega(c)$. It then follows that if the loss is sufficiently small, both $\mu_0$ and $\mu_1$ have correct sign, i.e. $\mu_0 > 0 > \mu_1$. (Otherwise, due to concentration of the output, the model makes wrong prediction and the loss is large). So we will eventually have $\delta = o(\mu_0)$ and $\delta = o(|\mu_1|)$.

## 5 DYNAMICS OF PARAMETERS

In this section, we describe the dynamics of each type of neurons through coordinate descent, which can be visualized in the following figure in which the arrows indicate movement between types that can happen with non-negligible probability.

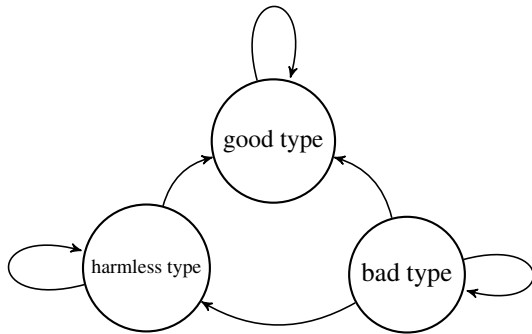

Figure 1: Dynamics of hidden neurons

There are two noteworthy points from this figure. First, "good type" parameters are preserved under coordinate descent. Second, there are no arrows coming into "bad type" except from itself.

These dynamics are proved by estimating the gradient with respect to the loss function for each type of neuron. Because of the non-linearity of the activation, we rely heavily on the concentration result proved above to get tight estimates. Without these concentration results, even estimating the sign of the gradient seems difficult. The proof and experiments about the dynamics of hidden neurons are deferred to the appendix.

## 6 LEARNING GUARANTEE

In this section, we prove our main result which states that with high probability a trained GCN can detect communities in SBM with any desired accuracy. The proof is based on the following theorem which shows that if $\mu_0$ and $\mu_1$ are separated enough, then the model achieves high accuracy.

**Theorem 6.1.** $\forall \epsilon > 0$, provided that the difference between $\mu_0$ and $\mu_1$ is large enough: $\sigma(-\frac{\mu_0 - \mu_1}{2}) < \frac{\epsilon}{2}$, if $\left|\mathbb{E}[\frac{\partial L}{\partial b_0}]\right| < \frac{\epsilon}{4}$, then $\mathbb{P}[\Delta < 0|\ell(x) = 0] < 4\epsilon, \mathbb{P}[\Delta > 0|\ell(x) = 1] < 4\epsilon$, where $\sigma(x) := 1/(1 + \exp(-x))$ represents the sigmoid function.

Next we show that the model can achieve such separation between $\mu_0$ and $\mu_1$ through coordinate descent. In order to make constant update of parameters at every epoch, we set an adaptive learning rate $\eta_k = \frac{1}{\mathbb{E}[Z^{(k)}]}$ where $Z^{(k)}$ is the value of $Z$ at the $k$-th epoch. We first refine Corollary 4.3 about the separation of output for each type of neuron $(m_0^i - m_1^i)$ using the dynamics of parameters.

**Theorem 6.2** (separation of output). *Let $m_0^i$ and $m_1^i$ be defined as in Section 4, train the model for $k$ epochs by the defined coordinate descent with adaptive learning rate $\eta_k = \frac{1}{\mathbb{E}[Z^{(k)}]}$,*

- *if the $i$-th neuron is of "good type", then*

$$m_0^i - m_1^i \geq A_i^{(0)} B_i^{(0)} \frac{\lambda}{2} \left(\frac{p-q}{p+q}\right)^2 \left(1 + \frac{\sqrt{2}\lambda}{8}\left(\frac{p-q}{p+q}\right)^2\right)^{2k}$$

- *if the $i$-th neuron is of "bad type", then*

$$m_0^i - m_1^i \geq -k\left(\left(A_i^{(0)}\right)^2 + \left(B_i^{(0)}\right)^2\right)\frac{\lambda p(p-q)}{(p+q)^2},$$

- *if the $i$-th neuron is of "harmless type", then*

$$m_0^i - m_1^i \geq 0,$$

where $A_i^{(0)} = \alpha_i^{(0)} - \alpha_i'^{(0)}$, $B_i^{(0)} = \beta_i^{(0)} - \beta_i'^{(0)}$.

Next we present a result about initialization, which shows that with high probability, there are enough "good type" neurons and parameters have appropriate scale.

**Lemma 6.3.** *Suppose all parameters in $W^{(0)}$ and $W^{(1)}$ are initialized independently following standard normal distribution. Then the number $h_g$ of neurons initialized as "good type" satisfies $\mathbb{P}[h_g \geq \frac{h}{8}] \geq 1 - \exp\left(-\frac{h}{64}\right)$. Furthermore,*

$$\mathbb{P}[\sum_{i=1}^{h}(\alpha_i - \alpha_i')^2 + (\beta_i - \beta_i')^2 \leq 5h] \geq 1 - O\left(\frac{1}{h}\right), \qquad \mathbb{P}[\sum_{\substack{\text{the } i\text{-th neuron} \\ \text{initialized as} \\ \text{"good type"}}} |\alpha_i - \alpha_i'||\beta_i - \beta_i'| \geq \frac{h}{80}] \geq 1 - O\left(\frac{1}{h}\right).$$

Now we can prove the final result.

*Proof of Theorem 2.2.* First we show that if the loss $\mathbb{E}[Z]$ is small enough, the model achieves desired accuracy. Indeed, if $\mathbb{E}[Z] < 2\epsilon$, since

$$\mathbb{E}[Z] = \mathbb{E}[Z|\text{pred is wrong}]\mathbb{P}[\text{pred is wrong}] + \mathbb{E}[Z|\text{pred is correct}]\mathbb{P}[\text{pred is correct}] \geq \frac{1}{2}\mathbb{P}[\text{pred is wrong}],$$

we have $\mathbb{P}[\text{pred is wrong}] \leq 4\epsilon$, i.e., $\mathbb{P}[\text{pred is correct}] > 1 - 4\epsilon$.

Otherwise, $\mathbb{E}[Z] \geq 2\epsilon$, since $\mathbb{E}[Z] = \frac{1}{2}(\mathbb{E}[Z|\ell(z) = 0] + \mathbb{E}[Z|\ell(z) = 1])$, we have $\mathbb{E}[Z|\ell(z) = 0] + \mathbb{E}[Z|\ell(z) = 1] \geq 4\epsilon$. On the other hand, $\left|\mathbb{E}[\frac{\partial L}{\partial b_0}]\right| < \epsilon$ implies that $\left|\mathbb{E}[Z|\ell(z) = 0] - \mathbb{E}[Z|\ell(z) = 1]\right| < 2\epsilon$.

By Theorem 6.2,

$$\mu_0 - \mu_1 = \sum_{i=1}^{h}(m_0^i - m_1^i) = \sum_{i \in \text{"good"}}(m_0^i - m_1^i) + \sum_{i \in \text{"bad"}}(m_0^i - m_1^i) + \sum_{i \in \text{"harmless"}}(m_0^i - m_1^i)$$

$$\geq \frac{\lambda}{2}\left(\frac{p-q}{p+q}\right)^2\left(1 + \frac{\sqrt{2}\lambda}{8}\left(\frac{p-q}{p+q}\right)^2\right)^{2k}\sum_{i \in \text{"good"}}A_i^{(0)}B_i^{(0)} - k\frac{\lambda p(p-q)}{(p+q)^2}\sum_{i \in \text{"bad"}}\left((A_i^{(0)})^2 + (B_i^{(0)})^2\right).$$

By Lemma 6.3, with probability $\geq 1 - O(\frac{1}{h})$,

$$\sum_{i \in \text{"good"}}A_i^{(0)}B_i^{(0)} \geq \frac{h}{80}, \qquad \sum_{i \in \text{"bad"}}\left((A_i^{(0)})^2 + (B_i^{(0)})^2\right) \leq 5h.$$

Since $h \geq \frac{1}{\delta}$, then with probability $\geq 1 - \delta$,

$$\mu_0 - \mu_1 \geq h\left(\frac{\lambda}{160}\left(\frac{p-q}{p+q}\right)^2\left(1 + \frac{\sqrt{2}\lambda}{8}\left(\frac{p-q}{p+q}\right)^2\right)^{2k} - k\frac{5\lambda p(p-q)}{(p+q)^2}\right)$$

$$\geq h(C_1(1 + C_2)^{2k} - C_3 k), \tag{4}$$

where $C_1$, $C_2$ and $C_3$ are constants determined by $p, q$ and $\lambda$.

By Theorem 6.1, if (4)$\geq 2\log\frac{2}{\epsilon}$ (then $\sigma(-\frac{\mu_0 - \mu_1}{2}) \leq \frac{\epsilon}{2}$), then the model achieves accuracy $\geq 1 - 4\epsilon$. It's sufficient to have

$$C_1(1 + C_2)^{2k} - C_3 k \geq 2\log\frac{2}{\epsilon},$$

i.e. $k = O(\log\log\frac{1}{\epsilon})$. $\qquad\square$

# 7  EXPERIMENTS

We show some experiments verifying Theorem 2.2. In particular, our experiments demonstrate that accuracy increases with $n$, the probability of high-accuracy models increases with $h$, and coordinate descent is able to recover high-accuracy models in the sparse regime of Assumption 2.1. Additional plots demonstrating the dynamics of hidden neurons with their ratios and differences can be seen in the appendix.

**Experiment 1** In this experiment, we plot the an estimate of the accuracy versus epoch for varying $n$. The parameters $p, q$ of SBM follow Assumption 2.1, where we choose $a = 1.0$ and $b = 0.7$. We set $h = 20, \lambda = 0.3$ and run 40 independent experiments for $n = 250, 500$ and 1000 respectively. In each experiment we train the model for 100 epochs. The training set has 40 randomly generated graphs from $\text{SBM}(n, p, q)$. We validate the performance by the percentage of correct predictions on 200 random vertices, each from a randomly generated graph. The result is shown Figure 2. The shaded region for each $n$ is obtained from the max, min and mean percentage of the 40 experiments. The result verifies Theorem 2.2 which shows that the accuracy of the model increases with $n$.

**Experiment 2** In this experiment, we show the effect of the number of hidden neurons $h$. The parameters of SBM are the same as Experiment 1. We set $h = 2, 5, 20$. For each pair of $(n, h)$ we run 40 independent experiments and show the distribution of validation in Figure 3. From the top row to the bottom, $n$ increases from 250 to 1000. From the left column to the right, $h$ increases from 2 to 20. In each plot, the x-axis represents the accuracy, while y-axis represents the count of experiments. According to Theorem 2.2, the probability of achieving high accuracy is $1 - O(1/h)$ and the accuracy increases with $n$. We can see that in each row of Figure 3, as $h$ increases, we have lager probability to achieve high accuracy; in each column, as $n$ increases, the model achieves higher accuracy. The results verify our theory in the paper.

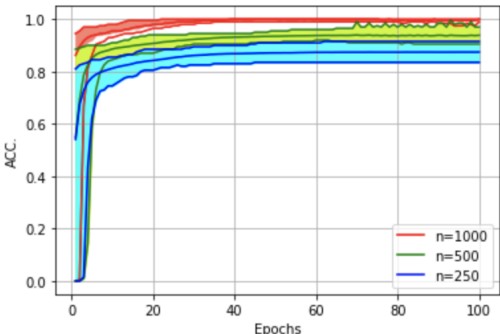

Figure 2: Display of accuracy versus epoch for varying $n$. Accuracy is computed on vertices drawn from random SBM graphs. The accuracy increases with epoch, and we also see that the lower bound on the accuracy increases with $n$, as expected from our theory.

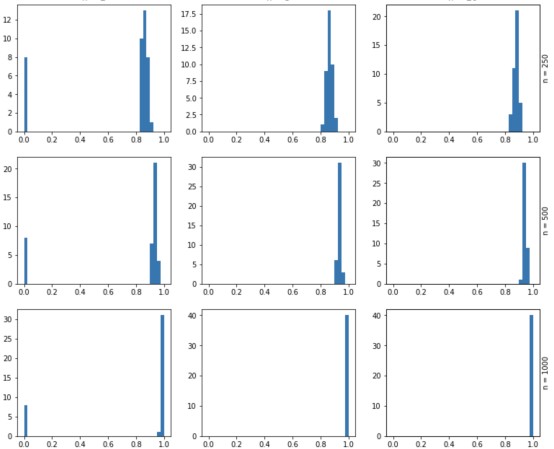

Figure 3: Display of accuracy for varying $n$ and $h$. For each row, $n$ is set to be 250 (top), 500 (middle), 1000 (bottom); for each column, $h$ is set to be 2 (left), 5 (middle), 20 (right). Probability of failure decreases as $h$ increases, and accuracy increases with $n$.

## 8 FUTURE DIRECTIONS

Graph neural networks offer a promising setting for progress on the more general theory of deep learning, because random graph models more plausibly capture the structure of real-world data compared to, e.g., the Gaussian inputs often used to prove deep learning guarantees for traditional feed-forward neural networks. This paper has initiated the project of proving training guarantees for semi-supervised learning using GCNs on SBM models, but much more work remains to be done. Arguably the sparsest SBM models (expected constant degree) are the most compelling from the perspective of modeling real-world communities, so it would be interesting to extend these results to that setting. Models with more than two blocks, or overlapping communities (Petti & Vempala, 2018) would be even closer to real-world structure. We hope this initial step spurs further interest in provable guarantees for training neural networks using plausible models of real-world data as the input distribution.

## 9 ACKNOWLEDGEMENT

This research was supported in part by NSF Grant 1849796. The author would like to thank Prof. John Wilmes for guiding the research, and also thank Prof. Tyler Maunu and Prof. Pengyu Hong for making the results more generalized and designing the experiments.

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

## A    CONCENTRATION AND SEPARATION OF OUTPUT

Let $N(x,0), N(x,1)$ denote the neighborhood of vertex $x$ with label 0 and 1 respectively, i.e. $N(x,0) = \{y \in G, y \sim x, \ell(y) = 0\}, N(x,1) = \{y \in G, y \sim x, \ell(y) = 1\}$. By the definition of SBM, both $|N(x,0)|$ and $|N(x,1)|$ are binomial random variables. For $\ell(x) = 0, |N(x,0)| \sim B(\frac{n}{2}, p), |N(x,1)| \sim B(\frac{n}{2}, q)$ and for $\ell(x) = 1, |N(x,0)| \sim B(\frac{n}{2}, q), |N(x,1)| \sim B(\frac{n}{2}, p)$. Moreover, $t_0^y$ and $t_1^y$ are also binomial random variables, for $\ell(y) = 0, t_0^y \sim B(\frac{n\lambda}{2}, p), t_1^y \sim B(\frac{n\lambda}{2}, q)$, similarly for $\ell(y) = 1$. Our following analysis is based on the condition that $|N(x,0)|, |N(x,1)|, t_0^x$ and $t_1^x$ are in their high probability range for all $x \in G$. Specifically we require the condition that for all $\ell(x) = 0$, (similar conditions for $\ell(x) = 1$ are omitted):

$$\left| |N(x,0)| - \frac{np}{2} \right| \leq O(np)^{\frac{5}{6}}, \left| |N(x,1)| - \frac{nq}{2} \right| \leq O(nq)^{\frac{5}{6}}; \qquad \text{(Cond)}$$

$$\left| t_0^x - \frac{n\lambda p}{2} \right| \leq O(np)^{\frac{5}{6}}, \left| t_1^x - \frac{n\lambda q}{2} \right| \leq O(nq)^{\frac{5}{6}}.$$

By tail bound of binomial random variables and union bound, we have

$$\mathbb{P}[(Cond)] \geq 1 - \frac{1}{n^2}.$$

Under this condition, we show the concentration of $\Delta_i$ for each type.

A.1 "GOOD TYPE" NEURONS

For convenience, according to the activation pattern of $\phi(\alpha_i t_0^y + \alpha_i' t_1^y)$, we further divide $(G_2)$ into subcases $(G_{2,1})$, $(G_{2,2})$ and $(G_{2,3})$ by to the ratio of $\left|\frac{\alpha_i}{\alpha_i'}\right|$. For example, in $(G_1)$ and $(G_{2,1})$, $\phi(\alpha_i t_0^y + \alpha_i' t_1^y)$ is active for both $\ell(y) = 0$ and $\ell(y) = 1$; in $(G_{2,1})$, it is only active for $\ell(y) = 0$.

$$\left|\frac{\alpha_i}{\alpha_i'}\right| > \frac{p}{q}(1 + \log^{-\frac{1}{3}} n) \tag{$G_{2,1}$}$$

$$1 < \left|\frac{\alpha_i}{\alpha_i'}\right| < \frac{p}{q}(1 - \log^{-\frac{1}{3}} n) \tag{$G_{2,2}$}$$

$$\frac{p}{q}(1 - \log^{-\frac{1}{3}} n) \leq \left|\frac{\alpha_i}{\alpha_i'}\right| \leq \frac{p}{q}(1 + \log^{-\frac{1}{3}} n) \tag{$G_{2,3}$}$$

We have the following estimation of $\Delta_i$ in "good type".

**Theorem A.1** (concentration of output from "good type" neurons). *If the $i$-th neuron is of "good type", then*

- *in both $(G_1)$ and $(G_{2,1})$:*

$$\mathbb{P}[\left|\Delta_i - \frac{\lambda(\beta_i - \beta_i')}{(p+q)^2}[(p^2 + q^2)\alpha_i + 2pq\alpha_i']\right| \leq (\alpha_i - \alpha_i')(\beta_i - \beta_i')O(\log^{-\frac{1}{2}} n)|\ell(x) = 0]$$
$$\geq 1 - O(\frac{1}{n^2})$$

$$\mathbb{P}[\left|\Delta_i - \frac{\lambda(\beta_i - \beta_i')}{(p+q)^2}[2pq\alpha_i + (p^2 + q^2)\alpha_i']\right| \leq (\alpha_i - \alpha_i')(\beta_i - \beta_i')O(\log^{-\frac{1}{2}} n)|\ell(x) = 1]$$
$$\geq 1 - O(\frac{1}{n^2})$$

- *in $(G_{2,2})$:*

$$\mathbb{P}[\left|\Delta_i - \frac{\lambda(\beta_i - \beta_i')p}{(p+q)^2}(p\alpha_i + q\alpha_i')\right| \leq (\alpha_i - \alpha_i')(\beta_i - \beta_i')O(\log^{-\frac{1}{2}} n)|\ell(x) = 0]$$
$$\geq 1 - O(\frac{1}{n^2})$$

$$\mathbb{P}[\left|\Delta_i - \frac{\lambda(\beta_i - \beta_i')q}{(p+q)^2}(p\alpha_i + q\alpha_i')\right| \leq (\alpha_i - \alpha_i')(\beta_i - \beta_i')O(\log^{-\frac{1}{2}} n)|\ell(x) = 1]$$
$$\geq 1 - O(\frac{1}{n^2})$$

- *in $(G_{2,3})$:*

$$\mathbb{P}[\left|\Delta_i - \frac{\lambda(\beta_i - \beta_i')p}{(p+q)^2}(p\alpha_i + q\alpha_i')\right| \leq (\alpha_i - \alpha_i')(\beta_i - \beta_i')O(\log^{-\frac{1}{2}} n)|\ell(x) = 0]$$
$$\geq 1 - O(\frac{1}{n^2})$$

$$\mathbb{P}[\left|\Delta_i - \frac{\lambda(\beta_i - \beta_i')q}{(p+q)^2}(p\alpha_i + q\alpha_i')\right| \leq (\alpha_i - \alpha_i')(\beta_i - \beta_i')O(\log^{-\frac{1}{2}} n)|\ell(x) = 1]$$
$$\geq 1 - O(\frac{1}{n^2}).$$

*Proof.* We have $\Delta_i = (\beta_i - \beta_i') \sum_{y \in G} \mathbb{1}[y \sim x] \frac{4\phi(\alpha_i t_0^y + \alpha_i' t_1^y)}{n^2(p+q)^2}$. We apply the method of averaged bounded difference (Dubhashi & Panconesi, 2009) to estimate $\Delta_i$. In different parameter regimes, $\phi(\alpha_i t_0^y + \alpha_i' t_1^y)$ has different activation patterns.

In $(G_1)$ and $(G_{2,1})$, $\phi(\alpha_i t_0^y + \alpha_i' t_1^y)$ is active with probability $1 - O(\frac{1}{n^2})$ for both $\ell(y) = 0$ and $\ell(y) = 1$. For $\ell(x) = 0$, at first we estimate $\mathbb{E}[\Delta_i]$. By condition (Cond):

$$\left| \mathbb{E}[\Delta_i] - \frac{\lambda(\beta_i - \beta_i')}{(p+q)^2}[(p^2+q^2)\alpha_i + 2pq\alpha_i'] \right| \leq (\alpha_i - \alpha_i')(\beta_i - \beta_i')O(\log^{-1} n).$$

Let $Y_j = \frac{4\phi(\alpha_i t_0^{y_j} + \alpha_i' t_1^{y_j})}{n^2(p+q)^2}$, then $\Delta_i = (\beta_i - \beta_i')\sum_j Y_j$. Based on condition (Cond), $\left| Y_j - \frac{2\lambda(p\alpha_i + q\alpha_i')}{n(p+q)^2} \right| \leq (\alpha_i - \alpha_i')O(\log^{-\frac{7}{2}} n)$ for $\ell(y_j) = 0$. For any $a_k, a_k'$,

$$\left| \mathbb{E}[Y_k|Y_1, \cdots, Y_{k-1}, Y_k = a_k] - \mathbb{E}[Y_k|Y_1, \cdots, Y_{k-1}, Y_k = a_k'] \right| \leq (\alpha_i - \alpha_i')O(\log^{-\frac{7}{2}} n).$$

Moreover, when the number of vertices with revealed labels are fixed,

$$\left| \mathbb{E}[Y_j|Y_1, \cdots, Y_{k-1}, Y_k = a_k] - \mathbb{E}[Y_j|Y_1, \cdots, Y_{k-1}, Y_k = a_k'] \right| \leq (\alpha_i - \alpha_i')O(\log^{-6} n),$$

for $j \geq k$. By condition (Cond), there are at most $O(\log^3 n)$ non-zero terms for $Y_k, 1 \leq k \leq n$. So

$$\left| \mathbb{E}[\sum_j Y_j|Y_1, \cdots, Y_{k-1}, Y_k = a_k] - \mathbb{E}[\sum_j Y_j|Y_1, \cdots, Y_{k-1}, Y_k = a_k'] \right| \leq (\alpha_i - \alpha_i')O(\log^{-\frac{7}{2}} n),$$

for $1 \leq k \leq n$. By the method of averaged bounded difference, we have

$$\mathbb{P}[\left| \Delta_i - \frac{\lambda(\beta_i - \beta_i')}{(p+q)^2}[(p^2+q^2)\alpha_i + 2pq\alpha_i'] \right| \leq (\alpha_i - \alpha_i')(\beta_i - \beta_i')O(\log n)^{-\frac{1}{2}} |\ell(x) = 0] \geq 1 - O(\frac{1}{n^2}).$$

Other regimes can be proved similarly. $\qquad\square$

## A.2 "BAD TYPE" NEURONS

For convenience of our analysis, we further divide $(B_2)$ into subcases $(B_{2,1})$, $(B_{2,2})$ and $(B_{2,3})$ according to the ratio of $\left| \frac{\alpha_i}{\alpha_i'} \right|$

$$\left| \frac{\alpha_i}{\alpha_i'} \right| > \frac{p}{q}(1 + \log^{-\frac{1}{3}} n) \tag{$B_{2,1}$}$$

$$\left| \frac{\alpha_i}{\alpha_i'} \right| \in (\frac{q}{p}(1 + \log^{-\frac{1}{3}} n), \frac{p}{q}(1 - \log^{-\frac{1}{3}} n)] \tag{$B_{2,2}$}$$

$$\left| \frac{\alpha_i}{\alpha_i'} \right| \in (\frac{p}{q}(1 - \log^{-\frac{1}{3}} n), \frac{p}{q}(1 + \log^{-\frac{1}{3}} n)] \tag{$B_{2,3}$}$$

We have the following estimation of $\Delta_i$ in "bad type".

**Theorem A.2** (concentration of output from "bad type" neurons)**.** *If the $i$-th neuron is of "bad type", we have:*

- *in $(B_1)$ and $(B_{2,1})$:*

$$\mathbb{P}[\left| \Delta_i - \frac{\lambda(\beta_i - \beta_i')}{(p+q)^2}[(p^2+q^2)\alpha_i + 2pq\alpha_i'] \right| \leq |\alpha_i - \alpha_i'||\beta_i - \beta_i'|O(\log^{-\frac{1}{2}} n)|\ell(x) = 0]$$

$$\geq 1 - O(\frac{1}{n^2})$$

$$\mathbb{P}[\left| \Delta_i - \frac{\lambda(\beta_i - \beta_i')}{(p+q)^2}[2pq\alpha_i + (p^2+q^2)\alpha_i'] \right| \leq |\alpha_i - \alpha_i'||\beta_i - \beta_i'|O(\log^{-\frac{1}{2}} n)|\ell(x) = 1]$$

$$\geq 1 - O(\frac{1}{n^2})$$

- *in ($B_{2,2}$):*

$$\mathbb{P}[\big|\Delta_i - \frac{\lambda(\beta_i - \beta_i')p}{(p+q)^2}(p\alpha_i + q\alpha_i')\big| \le |\alpha_i - \alpha_i'||\beta_i - \beta_i'|O(\log^{-\frac{1}{2}} n)|\ell(x) = 0]$$

$$\ge 1 - O(\frac{1}{n^2})$$

$$\mathbb{P}[\big|\Delta_i - \frac{\lambda(\beta_i - \beta_i')q}{(p+q)^2}(p\alpha_i + q\alpha_i')\big| \le |\alpha_i - \alpha_i'||\beta_i - \beta_i'|O(\log^{-\frac{1}{2}} n)|\ell(x) = 1]$$

$$\ge 1 - O(\frac{1}{n^2})$$

- *in ($B_{2,3}$):*

$$\mathbb{P}[\big|\Delta_i - \frac{\lambda(\beta_i - \beta_i')p}{(p+q)^2}(p\alpha_i + q\alpha_i')\big| \le |\alpha_i - \alpha_i'||\beta_i - \beta_i'|O(\log^{-\frac{1}{2}} n)|\ell(x) = 0]$$

$$\ge 1 - O(\frac{1}{n^2})$$

$$\mathbb{P}[\big|\Delta_i - \frac{\lambda(\beta_i - \beta_i')q}{(p+q)^2}(p\alpha_i + q\alpha_i')\big| \le |\alpha_i - \alpha_i'||\beta_i - \beta_i'|O(\log^{-\frac{1}{2}} n)|\ell(x) = 1]$$

$$\ge 1 - O(\frac{1}{n^2}).$$

- *in ($B_3$):*

$$\mathbb{P}[\big|\Delta_i\big| \le |\alpha_i - \alpha_i'||\beta_i - \beta_i'|O(\log^{-\frac{1}{2}} n)|\ell(x) = 0 \text{ or } 1] \ge 1 - O(\frac{1}{n^2}).$$

*Proof.* The proof is similar as Theorem A.1 $\qquad\qquad\square$

## A.3 "HARMLESS TYPE" NEURONS

We have the following estimation of $\Delta_i$ in "harmless type".

**Theorem A.3** (concentration of output from "harmless type" neurons)**.** *If the $i$-th neuron is of "harmless type", we have:*

- *in ($H_1$):*

$$\mathbb{P}[\big|\Delta_i - \frac{\lambda(\beta_i - \beta_i')p}{(p+q)^2}(p\alpha_i + q\alpha_i')\big| \le (\alpha_i - \alpha_i')(\beta_i - \beta_i')O(\log^{-\frac{1}{2}} n)|\ell(x) = 0]$$

$$\ge 1 - O(\frac{1}{n^2})$$

$$\mathbb{P}[\big|\Delta_i - \frac{\lambda(\beta_i - \beta_i')q}{(p+q)^2}(p\alpha_i + q\alpha_i')\big| \le (\alpha_i - \alpha_i')(\beta_i - \beta_i')O(\log^{-\frac{1}{2}} n)|\ell(x) = 1]$$

$$\ge 1 - O(\frac{1}{n^2})$$

- *in ($H_2$): $\Delta_i = 0$ for both $\ell(x) = 0$ and 1.*

## A.4 SEPARATION OF OUTPUT

Previous subsections have shown the concentration of $\Delta_i$ for each type of neurons. For the $i$-th neuron, we write the concentrated value as $m_0^i$ if $\ell(x) = 0$ and $m_1^i$ if $\ell(x) = 1$. From Theorem A.1, A.2 and A.3, we have the following result about the value of $m_0^i - m_1^i$ by straightforward computation.

**Corollary A.4.** *We have the following result about $m_0^i - m_1^i$ for $1 \leq i \leq h$:*

- *if the $i$-the neuron is of "good type":*

$$\text{in } (G_1) \text{ and } (G_{2,1}) : m_0^i - m_1^i = \lambda|\alpha_i - \alpha_i'||\beta_i - \beta_i'|\left(\frac{p-q}{p+q}\right)^2$$

$$\text{in } (G_{2,2}) : m_0^i - m_1^i = \lambda|\beta_i - \beta_i'||p\alpha_i + q\alpha_i'|\frac{p-q}{(p+q)^2}$$

$$\geq \frac{\lambda}{2}|\alpha_i - \alpha_i'||\beta_i - \beta_i'|\left(\frac{p-q}{p+q}\right)^2$$

$$\text{in } (G_{2,3}) : m_0^i - m_1^i = \lambda|\beta_i - \beta_i'||p\alpha_i + q\alpha_i'|\frac{p-q}{(p+q)^2}$$

$$\geq \lambda|\alpha_i - \alpha_i'||\beta_i - \beta_i'|\frac{(p-q)\Lambda_3}{(p+q)^2},$$

*where $\Lambda_3 = \frac{(1-\log^{-\frac{1}{3}} n)p^2 - q^2}{(1-\log^{-\frac{1}{3}} n)p+q}$.*

- *if the $i$-the neuron is of "bad type":*

$$\text{in } (B_1) \text{ and } (B_{2,1}) : m_0^i - m_1^i = -\lambda|\alpha_i - \alpha_i'||\beta_i - \beta_i'|\left(\frac{p-q}{p+q}\right)^2$$

$$\text{in } (B_{2,2}) : m_0^i - m_1^i = -\lambda|\beta_i - \beta_i'||p\alpha_i + q\alpha_i'|\frac{p-q}{(p+q)^2}$$

$$\geq -\lambda|\alpha_i - \alpha_i'||\beta_i - \beta_i'|\frac{(p-q)\Lambda_3}{(p+q)^2}$$

$$\text{in } (B_{2,3}) : m_0^i - m_1^i = -\lambda|\beta_i - \beta_i'||p\alpha_i + q\alpha_i'|\frac{p-q}{(p+q)^2}$$

$$\geq -\lambda|\alpha_i - \alpha_i'||\beta_i - \beta_i'|\frac{(p-q)\Lambda_1}{(p+q)^2}$$

$$\text{in } (B_3) : m_0^i - m_1^i = 0,$$

*where $\Lambda_1 = \frac{(1+\log^{-\frac{1}{3}} n)p^2 - q^2}{(1+\log^{-\frac{1}{3}} n)p+q}, \Lambda_3 = \frac{(1-\log^{-\frac{1}{3}} n)p^2 - q^2}{(1-\log^{-\frac{1}{3}} n)p+q}$.*

- *if the $i$-the neuron is of "harmless type":*

$$\text{in } (H_1) : m_0^i - m_1^i = \lambda|\beta_i - \beta_i'||p\alpha_i + q\alpha_i'|\frac{p-q}{(p+q)^2}$$

$$\geq \lambda|\alpha_i - \alpha_i'||\beta_i - \beta_i'|\frac{(p-q)\Lambda_5}{(p+q)^2}$$

$$\text{in } (H_2) : m_0^i - m_1^i = 0,$$

*where $\Lambda_5 = \frac{pq\log^{-\frac{1}{3}}}{p+(1+\log^{-\frac{1}{3}})q}$.*

## B   DYNAMICS OF PARAMETERS

We consider the cross-entropy loss in training. The loss on a particular vertex $x$ is

$$L(x) = -\log O_{\ell(x)}(x),$$

where $O_0(x)$ and $O_1(x)$ are the first and second component of the output respectively, i.e.

$$O_0(x) = \frac{\exp(g_0(x))}{\exp(g_0(x)) + \exp(g_1(x))}, \quad O_1(x) = \frac{\exp(g_1(x))}{\exp(g_0(x)) + \exp(g_1(x))}.$$

For a given graph $G$ generated by SBM, we set the objective function $L(G)$ as the average loss over all the vertices with revealed labels[2], i.e.

$$L(G) = \frac{1}{\#\{x \in G : \ell(x) \text{ is revealed}\}} \sum_{x:\ell(x) \text{ revealed}} L(x).$$

We first show the partial derivatives of parameters.

**Theorem B.1** (derivatives of parameters). *For $1 \leq i \leq h$, let $x$ be a vertex, $\ell(x)$ its true label, $L(x) = -\log O_{\ell(x)}(x)$, then*

$$\frac{\partial L}{\partial \alpha_i} = \frac{4}{n^2(p+q)^2}(\beta_i - \beta_i')Z(-1)^{1-\ell(x)} \sum_y \mathbb{1}[y \sim x]\mathbb{1}[\alpha_i t_0^y + \alpha_i' t_1^y \geq 0]t_0^y$$

$$\frac{\partial L}{\partial \alpha_i'} = \frac{4}{n^2(p+q)^2}(\beta_i - \beta_i')Z(-1)^{1-\ell(x)} \sum_y \mathbb{1}[y \sim x]\mathbb{1}[\alpha_i t_0^y + \alpha_i' t_1^y \geq 0]t_1^y$$

$$\frac{\partial L}{\partial \beta_i} = \frac{4}{n^2(p+q)^2}Z(-1)^{1-\ell(x)} \sum_y \mathbb{1}[y \sim x]\phi(\alpha_i t_0^y + \alpha_i' t_1^y)$$

$$\frac{\partial L}{\partial \beta_i'} = -\frac{4}{n^2(p+q)^2}Z(-1)^{1-\ell(x)} \sum_y \mathbb{1}[y \sim x]\phi(\alpha_i t_0^y + \alpha_i' t_1^y)$$

$$\frac{\partial L}{\partial b_0} = (-1)^{1-\ell(x)}Z$$

$$\frac{\partial L}{\partial b_1} = (-1)^{\ell(x)}Z,$$

*where $Z = \frac{\exp(g_{1-\ell(x)}(x))}{\exp(g_0(x))+\exp(g_1(x))}, t_0^y$ and $t_1^y$ are the numbers of neighbors of $y$(including perhaps $y$ itself) with revealed labels in class 0 and class 1 respectively.*

*Proof.* We compute $\frac{\partial L}{\partial \alpha_i}, \frac{\partial L}{\partial \beta_i}$ and $\frac{\partial L}{\partial b_0}$, others can be computed symmetrically. We have

$$L(x) = -\log O_{\ell(x)}(x) = \log(\exp(g_0(x)) + \exp(g_1(x))) - g_{\ell(x)}(x),$$

since $O_j(x) = \frac{\exp(g_j(x))}{\exp(g_0(x))+\exp(g_1(x))}, j = 0, 1$. So

$$\frac{\partial L}{\partial \alpha_i} = \frac{e^{g_0(x)}\frac{\partial g_0(x)}{\partial \alpha_i} + e^{g_1(x)}\frac{\partial g_1(x)}{\partial \alpha_i}}{e^{g_0(x)} + e^{g_1(x)}} - \frac{\partial g_{\ell(x)}(x)}{\partial \alpha_i}$$

$$= (-1)^{1-\ell(x)}Z\left(\frac{\partial g_0(x)}{\partial \alpha_i} - \frac{\partial g_1(x)}{\partial \alpha_i}\right).$$

Since $g_j(x) = f_j(x) + b_j, \frac{\partial g_j(x)}{\partial \alpha_i} = \frac{\partial f_j(x)}{\partial \alpha_i}, j = 0, 1$. By (2)

$$\frac{\partial f_0(x)}{\partial \alpha_i} = \frac{4}{n^2(p+q)^2} \sum_y \mathbb{1}[y \sim x]\beta_i\mathbb{1}[\alpha_i t_0^y + \alpha_i' t_1^y \geq 0]t_0^y$$

$$\frac{\partial f_1(x)}{\partial \alpha_i} = \frac{4}{n^2(p+q)^2} \sum_y \mathbb{1}[y \sim x]\beta_i'\mathbb{1}[\alpha_i t_0^y + \alpha_i' t_1^y \geq 0]t_0^y.$$

Therefore

$$\frac{\partial L}{\partial \alpha_i} = \frac{4}{n^2(p+q)^2}(-1)^{1-\ell(x)}Z(\beta_i - \beta_i') \sum_y \mathbb{1}[y \sim x]\mathbb{1}[\alpha_i t_0^y + \alpha_i' t_1^y \geq 0]t_0^y.$$

---

[2]We abuse the notation $L$ for $L(x)$ and $L(G)$, but the meaning is clear from the context.

Next we compute $\frac{\partial L}{\partial \beta_i}$. Similar as above, $\frac{\partial L}{\partial \beta_i} = (-1)^{1-\ell(x)} Z \big( \frac{\partial f_0(x)}{\partial \beta_i} - \frac{\partial f_1(x)}{\partial \beta_i} \big)$. By (2)

$$\frac{\partial f_0(x)}{\partial \beta_i} = \frac{4}{n^2(p+q)^2} \sum_y \mathbb{1}[y \sim x] \phi(\alpha_i t_0^y + \alpha_i' t_1^y)$$

$$\frac{\partial f_1(x)}{\partial \alpha_i} = 0.$$

So

$$\frac{\partial L}{\partial \beta_i} = \frac{4}{n^2(p+q)^2} (-1)^{1-\ell(x)} Z \sum_y \mathbb{1}[y \sim x] \phi(\alpha_i t_0^y + \alpha_i' t_1^y).$$

Lastly,

$$\frac{\partial L}{\partial b_0} = (-1)^{1-\ell(x)} Z \left( \frac{\partial g_0(x)}{\partial b_0} - \frac{\partial g_1(x)}{\partial b_0} \right)$$

$$= (-1)^{1-\ell(x)} Z,$$

since $\frac{\partial g_0(x)}{\partial b_0} = 1, \frac{\partial g_1(x)}{\partial b_0} = 0$. □

In the following, we will use Theorem B.1 to analyze the dynamics of neurons of each type. As we can see, all of $\frac{\partial L}{\partial \alpha_i}, \frac{\partial L}{\partial \alpha_i'}, \frac{\partial L}{\partial \beta_i}$ and $\frac{\partial L}{\partial \beta_i'}$ have the form $YZ$. In order to estimate these derivatives, we show the concentration of $Y$ and $Z$ respectively. To estimate the concentration of $Z$, we need the concentration of output obtained in Section 4. For any $\epsilon > 0$, we require the probability of concentration in (3) to be at least $1 - \tilde{\epsilon}$, where $\tilde{\epsilon} = o(\epsilon)$. In particular, if we choose $\tilde{\epsilon} = \epsilon^2$, then we set $1 - O(\frac{1}{n}) \geq 1 - \epsilon^2$, i.e.

$$n \geq \Omega\left(\frac{1}{\epsilon}\right)^2. \tag{5}$$

Our following analysis will be based on this condition.

Meanwhile in order to have balanced performance in each epoch of coordinate descent, we require $\left| \mathbb{E}[\frac{\partial L}{\partial b_0}] \right| < \frac{\tilde{\epsilon}}{2}$. Since $\mathbb{E}[\frac{\partial L}{\partial b_0}] = \frac{1}{2}(-\mathbb{E}[Z|\ell(x) = 0] + \mathbb{E}[Z|\ell(x) = 1]))$, we have

$$\left| \mathbb{E}[Z|\ell(x) = 0] - \mathbb{E}[Z|\ell(x) = 1] \right| < \tilde{\epsilon}. \tag{6}$$

We have the following relation between $\mu_0$ and $\mu_1$. In the following, $\sigma$ represents the sigmoid function: $\sigma(x) = \frac{1}{1+e^{-x}}$.

**Proposition B.2.** *If $\left| \mathbb{E}[Z|\ell(x) = 0] - \mathbb{E}[Z|\ell(x) = 1] \right| < \tilde{\epsilon}$, then $|\sigma(-\mu_0) - \sigma(\mu_1)| \leq \sigma'(\mu_0 - \delta)\delta + \sigma'(\mu_1 + \delta)\delta + 3\tilde{\epsilon}$, where $\delta$ is as shown in Corollary 4.2.*

*Proof.* We have

$$Z = \begin{cases} \sigma(-\Delta), & \ell(x) = 0 \\ \sigma(\Delta), & \ell(x) = 1 \end{cases}$$

For $\ell(x) = 0$, by Lagrange mean value theorem, $|\sigma(-\mu_0) - Z| = |\sigma(-\mu_0) - \sigma(-\Delta)| = \sigma'(\xi)(\Delta - \mu_0)$, where $\xi$ is between $-\mu_0$ and $-\Delta$. By Corollary 4.2 and the condition of $n$, $|\Delta - \mu_0| \leq \delta$ with probability $\geq 1 - \tilde{\epsilon}$. From the remark following Corollary A.4, we have $\sigma'(\xi) \leq \sigma'(-\mu_0 + \delta) = \sigma'(\mu_0 - \delta)$,[3] with probability $\geq 1 - \tilde{\epsilon}$. Then we have

$$\mathbb{P}[|\sigma(-\mu_0) - Z| \leq \sigma'(\mu_0 - \delta)\delta|\ell(x) = 0] \geq 1 - \tilde{\epsilon}.$$

Since

$$\mathbb{E}[Z|\ell(x) = 0] = \mathbb{E}[Z||\sigma(-\mu_0) - Z| \leq \sigma'(\mu_0 - \delta)\delta]\mathbb{P}[|\sigma(-\mu_0) - Z| \leq \sigma'(\mu_0 - \delta)\delta]$$
$$+ \mathbb{E}[Z||\sigma(-\mu_0) - Z| > \sigma'(\mu_0 - \delta)\delta]\mathbb{P}[|\sigma(-\mu_0) - Z| > \sigma'(\mu_0 - \delta)\delta],$$

---

[3] $\sigma'(x)$ is even.

then (note that $0 < Z < 1$)

$$\mathbb{E}[Z|\ell(x) = 0] \leq \sigma(-\mu_0) + \sigma'(\mu_0 - \delta)\delta + \tilde{\epsilon}$$

and

$$\mathbb{E}[Z|\ell(x) = 0] \geq (\sigma(-\mu_0) - \sigma'(\mu_0 - \delta)\delta)(1 - \tilde{\epsilon}),$$

i.e.

$$\mathbb{E}[Z|\ell(x) = 0] - \sigma(-\mu_0) \leq \sigma'(\mu_0 - \delta)\delta + \tilde{\epsilon}$$

and

$$\begin{aligned}
\mathbb{E}[Z|\ell(x) = 0] - \sigma(-\mu_0) &\geq -\sigma'(\mu_0 - \delta)\delta - \tilde{\epsilon}(\sigma(-\mu_0) - \sigma'(\mu_0 - \delta)\delta) \\
&= -\sigma'(\mu_0 - \delta)\delta(1 - \tilde{\epsilon}) - \tilde{\epsilon}\sigma(-\mu_0) \\
&\geq -\sigma'(\mu_0 - \delta)\delta - \tilde{\epsilon}.
\end{aligned}$$

So

$$\big|\mathbb{E}[Z|\ell(x) = 0] - \sigma(-\mu_0)\big| \leq \sigma'(\mu_0 - \delta)\delta + \tilde{\epsilon}.$$

Similarly

$$\big|\mathbb{E}[Z|\ell(x) = 1] - \sigma(\mu_1)\big| \leq \sigma'(\mu_1 + \delta)\delta + \tilde{\epsilon}.$$

By triangle inequality,

$$\begin{aligned}
\big|\sigma(-\mu_0) - \sigma(\mu_1)\big| &\leq \big|\sigma(-\mu_0) - \mathbb{E}[Z|\ell(x) = 0]\big| + \big|\mathbb{E}[Z|\ell(x) = 0] - \mathbb{E}[Z|\ell(x) = 1]\big| \\
&\quad + \big|\mathbb{E}[Z|\ell(x) = 1] - \sigma(\mu_1)\big| \\
&\leq \sigma'(\mu_0 - \delta)\delta + \sigma'(\mu_1 + \delta)\delta + 3\tilde{\epsilon}.
\end{aligned}$$

$\square$

From the proof above, we can directly obtain the following corollary about $Z$.

**Corollary B.3.**

$$\mathbb{P}\big[\big|Z - \mathbb{E}[Z|\ell(x) = 0]\big| \leq 2\sigma'(\mu_0 - \delta)\delta + \tilde{\epsilon}\big|\ell(x) = 0\big] \geq 1 - \tilde{\epsilon}$$
$$\mathbb{P}\big[\big|Z - \mathbb{E}[Z|\ell(x) = 1]\big| \leq 2\sigma'(\mu_1 + \delta)\delta + \tilde{\epsilon}\big|\ell(x) = 1\big] \geq 1 - \tilde{\epsilon}.$$

In order to obtain the concentration of $Z$, we need to estimate $\sigma'(\mu_0 - \delta)\delta$ and $\sigma'(\mu_1 + \delta)\delta$. The following proposition is based on the condition that $|\mu_0 + \mu_1| \geq 4\delta$. If $|\mu_0 + \mu_1| < 4\delta$, set $c := \mu_0 - \mu_1$, we have $\mu_0 > \frac{c}{2} - 2\delta$ and $\mu_1 < -\frac{c}{2} + 2\delta$. Then the concentration of output shown in Corollary 4.2 can guarantee $1 - \epsilon$ accuracy of the model for any $\epsilon > 0$. In fact, from $|\Delta - \mu_0| < \delta$, we have $\Delta > \mu_0 - \delta > \frac{c}{2} - 3\delta > 0$, due to $\delta = o(c)$. So

$$\mathbb{P}[\Delta > 0|\ell(x) = 0] \geq \mathbb{P}[|\Delta - \mu_0| < \delta|\ell(x) = 0] \geq 1 - \tilde{\epsilon}.$$

Similarly,

$$\mathbb{P}[\Delta < 0|\ell(x) = 1] \geq \mathbb{P}[|\Delta - \mu_0| < \delta|\ell(x) = 1] \geq 1 - \tilde{\epsilon}.$$

Since $\tilde{\epsilon} = o(\epsilon)$, the model achieves overall accuracy $\geq 1 - \epsilon$.

**Proposition B.4.** *If $|\mu_0 + \mu_1| \geq 4\delta$, then $\sigma'(\mu_0 - \delta)\delta = O(\tilde{\epsilon}), \sigma'(\mu_1 + \delta)\delta = O(\tilde{\epsilon})$.*

*Proof.* First, we estimate the lower bound of $|\sigma(-\mu_0) - \sigma(\mu_1)|$ via the Fundamental Theorem of Calculus. We have $|\sigma(-\mu_0) - \sigma(\mu_1)| = \big|\int_{-\mu_0}^{\mu_1} \sigma'(t)\, dt\big|$.

If $-\mu_0 < \mu_1 < 0$, since $\mu_0 + \mu_1 \geq 4\delta$, we divide the interval $[-\mu_0, \mu_1]$ into $[-\mu_0, -\mu_0 + 2\delta] \cup [-\mu_0 + 2\delta, \mu_1 - 2\delta] \cup [\mu_1 - 2\delta, \mu_1]$ and estimate the lower bound of the integral. Since $\sigma'(x)$ is increasing on $(-\infty, 0]$, we have

$$\int_{-\mu_0}^{\mu_1} \sigma'(t)\, dt \geq \sigma'(-\mu_0) \cdot 2\delta + I_1 + \sigma'(\mu_1 - 2\delta) \cdot 2\delta, \tag{7}$$

where $I_1 = \int_{-\mu_0 + 2\delta}^{\mu_1 - 2\delta} \sigma'(t)\, dt$. If $\mu_1 < -\mu_0 < 0$, similarly we have

$$\int_{\mu_1}^{-\mu_0} \sigma'(t)\, dt \geq \sigma'(\mu_1) \cdot 2\delta + I_2 + \sigma'(-\mu_0 - 2\delta) \cdot 2\delta, \tag{8}$$

where $I_2 = \int_{\mu_1+2\delta}^{-\mu_0-2\delta} \sigma'(t)\, dt$. We have a uniform lower bound from (7) and (8):

$$\left| \int_{-\mu_0}^{\mu_1} \sigma'(t)\, dt \right| \geq \sigma'(-\mu_0 - 2\delta) \cdot 2\delta + \sigma'(\mu_1 - 2\delta) \cdot 2\delta + I, \tag{9}$$

where $I = \min\{I_1, I_2\}$.

Furthermore, by Proposition B.2,

$$|\sigma(-\mu_0) - \sigma(\mu_1)| \leq \sigma'(\mu_0 - \delta)\delta + \sigma'(\mu_1 + \delta)\delta + 3\tilde{\epsilon}. \tag{10}$$

Combine (9) and (10):

$$2\sigma'(-\mu_0 - 2\delta)\delta + 2\sigma'(\mu_1 - 2\delta)\delta \leq \sigma'(-\mu_0 + \delta)\delta + \sigma'(\mu_1 + \delta)\delta + 3\tilde{\epsilon}. \tag{11}$$

By Lagrange mean value theorem,

$$\sigma'(-\mu_0 - 2\delta) = \sigma'(-\mu_0 + \delta) - 3\sigma''(\xi_0)\delta$$
$$\sigma'(\mu_1 - 2\delta) = \sigma'(\mu_1 + \delta) - 3\sigma''(\xi_1)\delta,$$

where $\xi_0 \in (-\mu_0 - 2\delta, -\mu_0 + \delta), \xi_1 \in (\mu_1 - 2\delta, \mu_1 + \delta)$. Plug these into (11):

$$\sigma'(\mu_0 - \delta)\delta + \sigma'(\mu_1 + \delta)\delta - 6\delta^2(\sigma''(\xi_0) + \sigma''(\xi_1)) \leq 3\tilde{\epsilon}.$$

Since $\delta^2(\sigma''(\xi_0) + \sigma''(\xi_1)) = o(\sigma'(\mu_0 - \delta)\delta)$ and $o(\sigma'(\mu_1 + \delta)\delta)$, we have

$$\sigma'(\mu_0 - \delta)\delta = O(\tilde{\epsilon})$$
$$\sigma'(\mu_1 + \delta)\delta = O(\tilde{\epsilon}).$$

$\square$

Combine Proposition B.4 and Corollary B.3, we have the following concentration of $Z$.

**Proposition B.5.**

$$\mathbb{P}\big[\big|Z - \mathbb{E}[Z|\ell(x) = 0]\big| \leq O(\tilde{\epsilon})|\ell(x) = 0\big] \geq 1 - \tilde{\epsilon}$$
$$\mathbb{P}\big[\big|Z - \mathbb{E}[Z|\ell(x) = 1]\big| \leq O(\tilde{\epsilon})|\ell(x) = 1\big] \geq 1 - \tilde{\epsilon}.$$

Under the condition of balanced performance, we have the following corollary about the concentration of $Z$ independent of the label of $x$.

**Corollary B.6.** *If* $\big|\mathbb{E}[\frac{\partial L}{\partial b_0}]\big| \leq \frac{\tilde{\epsilon}}{2}$, *then* $\mathbb{P}\big[\big|Z - \mathbb{E}[Z]\big| \leq O(\tilde{\epsilon})\big] \geq 1 - \tilde{\epsilon}$.

*Proof.* Since $\mathbb{E}[\frac{\partial L}{\partial b_0}] = \frac{1}{2}(\mathbb{E}[Z|\ell(x) = 0] - \mathbb{E}[Z|\ell(x) = 1])$, we have

$$\big|\mathbb{E}[Z|\ell(x) = 0] - \mathbb{E}[Z|\ell(x) = 1]\big| \leq \tilde{\epsilon}.$$

On the other hand,

$$\mathbb{E}[Z] = \frac{1}{2}\big(\mathbb{E}[Z|\ell(x) = 0] + \mathbb{E}[Z|\ell(x) = 1]\big).$$

So we have $\big|\mathbb{E}[Z] - \mathbb{E}[Z|\ell(x) = 0]\big| \leq \frac{\tilde{\epsilon}}{2}$. By Proposition B.5, $\mathbb{P}[\big|Z - \mathbb{E}[Z]\big| \leq O(\tilde{\epsilon})] \geq 1 - \tilde{\epsilon}$. $\square$

Now we can derive the estimation of the derivatives.

**Theorem B.7** (concentration of derivatives). *For loss on the whole graph* $L = L(G)$, *with probability* $\geq 1 - O(\frac{1}{n})$, *we have* [4]

1. *If* $\alpha_i > \alpha_i' > 0$ *or* $\alpha_i > 0 > \alpha_i', |\frac{\alpha_i}{\alpha_i'}| \geq \frac{p}{q}(1 + \log^{-\frac{1}{3}} n)$, *then*

$$\left| \frac{\partial L}{\partial \alpha_i} + (\beta_i - \beta_i')\frac{\lambda}{2}\left(\frac{p-q}{p+q}\right)^2 \mathbb{E}[Z] \right| \leq |\beta_i - \beta_i'|\mathbb{E}[Z]O(\log^{-\frac{1}{2}} n) \tag{12}$$

$$\left| \frac{\partial L}{\partial \alpha_i'} - (\beta_i - \beta_i')\frac{\lambda}{2}\left(\frac{p-q}{p+q}\right)^2 \mathbb{E}[Z] \right| \leq |\beta_i - \beta_i'|\mathbb{E}[Z]O(\log^{-\frac{1}{2}} n) \tag{13}$$

$$\left| \frac{\partial L}{\partial \beta_i} + (\alpha_i - \alpha_i')\frac{\lambda}{2}\left(\frac{p-q}{p+q}\right)^2 \mathbb{E}[Z] \right| \leq |\alpha_i - \alpha_i'|\mathbb{E}[Z]O(\log^{-\frac{1}{2}} n). \tag{14}$$

---

[4]Since $\frac{\partial L}{\partial \beta_i'} = -\frac{\partial L}{\partial \beta_i}$ (see Theorem B.1), we only need to estimate $\frac{\partial L}{\partial \beta_i}$.

2. *If $\alpha_i > 0 > \alpha'_i, |\frac{\alpha_i}{\alpha'_i}| \in [\frac{q}{p}(1+\gamma), \frac{p}{q}(1-\log^{-\frac{1}{3}} n)]$, where $\gamma \in [\log^{-\frac{1}{3}} n, (\frac{p}{q})^2(1-\log^{-\frac{1}{3}} n) - 1]$, then*

$$\left| \frac{\partial L}{\partial \alpha_i} + (\beta_i - \beta'_i)\frac{\lambda p(p-q)}{2(p+q)^2}\mathbb{E}[Z] \right| \le |\beta_i - \beta'_i|\mathbb{E}[Z]O(\log^{-\frac{1}{2}} n) \tag{15}$$

$$\left| \frac{\partial L}{\partial \alpha'_i} + (\beta_i - \beta'_i)\frac{\lambda q(p-q)}{2(p+q)^2}\mathbb{E}[\mathbb{Z}] \right| \le |\beta_i - \beta'_i|\mathbb{E}[Z]O(\log^{-\frac{1}{2}} n) \tag{16}$$

$$\left| \frac{\partial L}{\partial \beta_i} + \frac{\lambda(p-q)(p\alpha_i + q\alpha'_i)}{2(p+q)^2}\mathbb{E}[Z] \right| \le |\alpha_i - \alpha'_i|\mathbb{E}[Z]O(\log^{-\frac{1}{2}} n). \tag{17}$$

3. *If $\alpha_i > 0 > \alpha'_i, |\frac{\alpha_i}{\alpha'_i}| \in (\frac{p}{q}(1-\log^{-\frac{1}{3}} n), \frac{p}{q}(1+\log^{-\frac{1}{3}} n))$, then*

$$\frac{\partial L}{\partial \beta_i} \in \left[ -(\alpha_i - \alpha'_i)\mathbb{E}[Z]\left( \frac{\lambda(p-q)\Lambda_1}{2(p+q)^2} + O(\log^{-\frac{1}{2}} n) \right), \right.$$
$$\left. -(\alpha_i - \alpha'_i)\mathbb{E}[Z]\left( \frac{\lambda(p-q)(\Lambda_3 - \Lambda_2)}{2(p+q)^2} - O(\log^{-\frac{1}{2}} n) \right) \right], \tag{18}$$

*where $\Lambda_1 = \frac{(1+\log^{-\frac{1}{3}} n)p^2 - q^2}{(1+\log^{-\frac{1}{3}} n)p+q}, \Lambda_2 = \frac{pq\log^{-\frac{1}{3}} n}{(1+\log^{-\frac{1}{3}} n)p+q}$ and $\Lambda_3 = \frac{(1-\log^{-\frac{1}{3}} n)p^2 - q^2}{(1-\log^{-\frac{1}{3}} n)p+q}$;*

- *if $\beta_i > \beta'_i$,*

$$\frac{\partial L}{\partial \alpha_i} \in \left[ -(\beta_i - \beta'_i)\mathbb{E}[Z]\left( \frac{\lambda p(p-q)}{2(p+q)^2} + O(\log^{-\frac{1}{2}} n) \right), \right.$$
$$\left. -(\beta_i - \beta'_i)\mathbb{E}[Z]\left( \frac{\lambda}{2}\left( \frac{p-q}{p+q} \right)^2 - O(\log^{-\frac{1}{2}} n) \right) \right] \tag{19}$$

$$\frac{\partial L}{\partial \alpha'_i} \in \left[ -(\beta_i - \beta'_i)\mathbb{E}[Z]\left( \frac{\lambda q(p-q)}{2(p+q)^2} + O(\log^{-\frac{1}{2}} n) \right), \right.$$
$$\left. (\beta_i - \beta'_i)\mathbb{E}[Z]\left( \frac{\lambda}{2}\left( \frac{p-q}{p+q} \right)^2 + O(\log^{-\frac{1}{2}} n) \right) \right] \tag{20}$$

$$\frac{\partial L}{\partial \alpha_i} - \frac{\partial L}{\partial \alpha'_i} \in \left[ -(\beta_i - \beta'_i)\mathbb{E}[Z]\left( \lambda\left( \frac{p-q}{p+q} \right)^2 + O(\log^{-\frac{1}{2}} n) \right), \right.$$
$$\left. -(\beta_i - \beta'_i)\mathbb{E}[Z]\left( \frac{\lambda}{2}\left( \frac{p-q}{p+q} \right)^2 - O(\log^{-\frac{1}{2}} n) \right) \right], \tag{21}$$

- *if $\beta_i \le \beta'_i$,*

$$\frac{\partial L}{\partial \alpha_i} \in \left[ -(\beta_i - \beta'_i)\mathbb{E}[Z]\left( \frac{\lambda}{2}\left( \frac{p-q}{p+q} \right)^2 - O(\log^{-\frac{1}{2}} n) \right), \right.$$
$$\left. -(\beta_i - \beta'_i)\mathbb{E}[Z]\left( \frac{\lambda p(p-q)}{2(p+q)^2} + O(\log^{-\frac{1}{2}} n) \right) \right] \tag{22}$$

$$\frac{\partial L}{\partial \alpha'_i} \in \left[ -(\beta_i - \beta'_i)\mathbb{E}[Z]\left( \frac{\lambda}{2}\left( \frac{p-q}{p+q} \right)^2 + O(\log^{-\frac{1}{2}} n) \right), \right.$$
$$\left. (\beta_i - \beta'_i)\mathbb{E}[Z]\left( \frac{\lambda q(p-q)}{2(p+q)^2} + O(\log^{-\frac{1}{2}} n) \right) \right] \tag{23}$$

$$\frac{\partial L}{\partial \alpha_i} - \frac{\partial L}{\partial \alpha'_i} \in \left[ -(\beta_i - \beta'_i)\mathbb{E}[Z]\left( \frac{\lambda}{2}\left( \frac{p-q}{p+q} \right)^2 - O(\log^{-\frac{1}{2}} n) \right), \right.$$
$$\left. -(\beta_i - \beta'_i)\mathbb{E}[Z]\left( \lambda\left( \frac{p-q}{p+q} \right)^2 + O(\log^{-\frac{1}{2}} n) \right) \right] \tag{24}$$

4. *If $\alpha_i > 0 > \alpha_i', |\frac{\alpha_i}{\alpha_i'}| \leq \frac{q}{p}(1 + \log^{-\frac{1}{3}} n), \beta_i < \beta_i'$, then*

$$\frac{\partial L}{\partial \alpha_i} - \frac{\partial L}{\partial \alpha_i'} \geq -|\beta_i - \beta_i'|O(\tilde{\epsilon}) \tag{25}$$

$$\frac{\partial L}{\partial \beta_i} \leq O(\tilde{\epsilon}). \tag{26}$$

*Proof.* We show the proof for item 1, other items can be proved similarly. Since $L(G)$ is the average of the losses over revealed vertices, we first show the concentration of $\frac{\partial L(x)}{\partial \alpha_i}$, then we show the concentration of $\frac{\partial L(G)}{\partial \alpha_i}$ using union bound. Since

$$\frac{\partial L(x)}{\partial \alpha_i} = (-1)^{1-\ell(x)} 4(\beta_i - \beta_i')Z \sum_y \frac{\mathbb{1}[y \sim x]\mathbb{1}[\alpha_i t_0^y + \alpha_i' t_1^y \geq 0]t_0^y}{n^2(p+q)^2},$$

we first show the concentration of $Y := (-1)^{1-\ell(x)} \sum_{y\sim x} \frac{4\mathbb{1}[\alpha_i t_0^y + \alpha_i' t_1^y \geq 0]t_0^y}{n^2(p+q)^2}$ using the method of averaged bounded difference. Similar as the proof of Theorem A.1, let $Y_j = (-1)^{1-\ell(x)}\frac{4\mathbb{1}[\alpha_i t_0^{y_j} + \alpha_i' t_1^{y_j} \geq 0]t_0^{y_j}}{n^2(p+q)^2}$. Based on Condition (Cond), for $\ell(x) = 0, |Y_j + \frac{2\lambda p}{n(p+q)^2}| \leq O(\log^{-\frac{7}{2}} n)$ for $\ell(y_j) = 0$. Similar results hold for $\ell(y_j) = 1, \ell(x) = 1$. So for any $a_k, a_k'$,

$$\left|\mathbb{E}[\sum_j Y_j|Y_1, \cdots, Y_{k-1}, Y_k = a_k] - \mathbb{E}[\sum_j Y_j|Y_1, \cdots, Y_{k-1}, Y_k = a_k']\right| \leq (\alpha_i - \alpha_i')O(\log^{-\frac{7}{2}} n).$$

By method of averaged bounded difference, for $\ell(x) = 0$,

$$\mathbb{P}\left[\left|\sum_{\ell(y_j)=0} Y_j + \lambda\left(\frac{p}{p+q}\right)^2\right| \leq O(\log^{-\frac{1}{2}} n)\right] \geq 1 - \exp\left(-2\log^3 n\right) \geq 1 - \frac{1}{n^2}.$$

Similarly

$$\mathbb{P}\left[\left|\sum_{\ell(y_j)=1} Y_j + \lambda\left(\frac{q}{p+q}\right)^2\right| \leq O(\log^{-\frac{1}{2}} n)\right] \geq 1 - \frac{1}{n^2}.$$

Hence

$$\mathbb{P}\left[\left|Y + \frac{\lambda(p^2+q^2)}{(p+q)^2}\right| \leq O(\log^{-\frac{1}{2}} n)\right] \geq 1 - \frac{1}{n^2}.$$

By Corollary B.6, $\mathbb{P}[|Z - \mathbb{E}[Z]| \leq O(\tilde{\epsilon})] \geq 1 - \tilde{\epsilon}$, so we have

$$\mathbb{P}\left[\left|\frac{\partial L(x)}{\partial \alpha_i} + (\beta_i - \beta_i')\lambda\frac{p^2+q^2}{(p+q)^2}\mathbb{E}[Z]\right| \leq |\beta_i - \beta_i'|\mathbb{E}[Z]O(\log^{-\frac{1}{2}} n)|\ell(x) = 0\right] \geq 1 - O\left(\frac{1}{n^2}\right).$$

For $\ell(x) = 1$, similarly we have

$$\mathbb{P}\left[\left|\frac{\partial L(x)}{\partial \alpha_i} - (\beta_i - \beta_i')\lambda\frac{2pq}{(p+q)^2}\mathbb{E}[Z]\right| \leq |\beta_i - \beta_i'|\mathbb{E}[Z]O(\log^{-\frac{1}{2}} n)|\ell(x) = 1\right] \geq 1 - O\left(\frac{1}{n^2}\right).$$

By union bound, we have (12). (13) and (14) can be proved similarly. □

Using Theorem B.7, we can analyze dynamics of neurons of each type. First, we introduce some notations. Let $\eta_k$ denote the learning rate at the $k$-th epoch, $Z^{(k)}$ be the value of $Z$ at the $k$-th epoch, $\alpha_i^{(k)}$ be the value of $\alpha_i$ at the $k$-th epoch, similar for $\alpha_i'^{(k)}, \beta_i^{(k)}$ and $\beta_i'^{(k)}$. In particular, $\alpha_i^{(0)}, \alpha_i'^{(0)}, \beta_i^{(0)}$ and $\beta_i'^{(0)}$ represent the values at initialization.

## B.1 "GOOD TYPE" NEURONS

In this section, we show that "good type" neurons stay in the "good type" regime throughout coordinate descent (Theorem B.8) using Theorem B.7.

**Theorem B.8.** *"Good type" neurons are preserved in the "good type" throughout coordinate descent with probability $\geq 1 - O(\frac{1}{n^2})$ over the SBM randomness.*

*Proof.* As shown in Section 4, "good type" regime is composed of $(G_1)$ and $(G_2)$, we show the dynamics of neurons in $(G_1)$ and $(G_2)$ respectively.

Assume that neuron $(\alpha_i^{(k)}, \alpha_i'^{(k)}, \beta_i^{(k)}, \beta_i'^{(k)})$ is in $(G_1)$, we show that it either stays in $(G_1)$ or moves into $(G_2)$ throughout coordinate descent. In fact, by (14), with probability $\geq 1 - O(\frac{1}{n^2})$, $\frac{\partial L}{\partial \beta_i^{(k)}} < 0 < \frac{\partial L}{\partial \beta_i'^{(k)}}$, so $\beta_i^{(k+1)} > \beta_i^{(k)}, \beta_i'^{(k+1)} < \beta_i'^{(k)}$ and hence $\beta_i^{(k+1)} - \beta_i'^{(k+1)} > \beta_i^{(k)} - \beta_i'^{(k)} > 0$. By (12) and (13), $\frac{\partial L}{\partial \alpha_i^{(k)}} < 0 < \frac{\partial L}{\partial \alpha_i'^{(k)}}$, so $\alpha_i^{(k+1)} > \alpha_i^{(k)}, \alpha_i'^{(k+1)} < \alpha_i'^{(k)}$. If $\alpha_i'^{(k+1)} > 0$, this neuron stays in $(G_1)$. If $\alpha_i'^{(k+1)} < 0$, since

$$\left| \frac{\alpha_i^{(k+1)}}{\alpha_i'^{(k+1)}} \right| = \left| \frac{\alpha_i^{(k)} - \eta_k \frac{\partial L}{\partial \alpha_i^{(k)}}}{\alpha_i'^{(k)} - \eta_k \frac{\partial L}{\partial \alpha_i'^{(k)}}} \right| > 1,$$

the neuron moves into $(G_2)$.

Assume that neuron is in $(G_2)$, we also show that it either moves into $(G_1)$ or stays in $(G_2)$. As shown in section 3.2, $(G_2) = (G_{2,1}) \cup (G_{2,2}) \cup (G_{2,3})$. If the neuron is in $(G_{2,1})$, again by (12), (13) and (14), $\alpha_i^{(k+1)} > \alpha_i^{(k)} > 0 > \alpha_i'^{(k)} > \alpha_i'^{(k+1)}, \left| \frac{\alpha_i^{(k+1)}}{\alpha_i'^{(k+1)}} \right| > 1, \beta_i^{(k+1)} > \beta_i^{(k)} > \beta_i'^{(k)} > \beta_i'^{(k+1)}$, so the neuron stays in $G_{2,2}$. If the neuron is in $G_{2,2}$, by (15), (16) and (17), $\frac{\partial L}{\partial \beta_i^{(k)}} < 0 < \frac{\partial L}{\partial \beta_i'^{(k)}}$, so $\beta_i^{(k+1)} > \beta_i'^{(k+1)}$. Also, $\frac{\partial L}{\partial \alpha_i^{(k)}} < \frac{\partial L}{\partial \alpha_i'^{(k)}} < 0$, so $\alpha_i^{(k+1)} > \alpha_i'^{(k+1)}, \left| \frac{\alpha_i^{(k+1)}}{\alpha_i'^{(k+1)}} \right| > \left| \frac{\alpha_i^{(k)}}{\alpha_i'^{(k)}} \right| > 1$. If $\alpha_i'^{(k+1)} < 0$, the neuron stays in $G_2$. If $\alpha_i'^{(k+1)} > 0$, it moves into $G_1$. If the neuron is in $G_{2,3}$, by (18) and (21), $\frac{\partial L}{\partial \beta_i^{(k)}} < 0 < \frac{\partial L}{\partial \beta_i'^{(k)}}, \frac{\partial L}{\partial \alpha_i^{(k)}} - \frac{\partial L}{\partial \alpha_i'^{(k)}} < 0$, so $\beta_i^{(k+1)} > \beta_i^{(k)} > \beta_i'^{(k)} > \beta_i'^{(k+1)}, \alpha_i^{(k+1)} - \alpha_i'^{(k+1)} > \alpha_i^{(k)} - \alpha_i'^{(k)} > 0$. By (19) and (20),

$$\frac{\partial L}{\partial \alpha_i} \leq -(\beta_i - \beta_i') \mathbb{E}[Z] \left( \frac{\lambda}{2} \left( \frac{p-q}{p+q} \right)^2 - O(\log^{-\frac{1}{2}} n) \right),$$

$$\frac{\partial L}{\partial \alpha_i'} \leq (\beta_i - \beta_i') \mathbb{E}[Z] \left( \frac{\lambda}{2} \left( \frac{p-q}{p+q} \right)^2 + O(\log^{-\frac{1}{2}} n) \right).$$

Similar as in $(G_{2,2})$, if $\alpha_i'^{(k+1)} < 0$, the neuron stays in $(G_2)$. If $\alpha_i'^{(k+1)} > 0$, it moves into $(G_1)$. $\square$

## B.2 "BAD TYPE" NEURONS

As shown in Section 4, neurons of "bad type" consist of two cases: $B_1$ and $B_2$, where $B_2 = B_{2,1} \cup B_{2,2} \cup B_{2,3} \cup B_3$. Since the output in $B_3$ is concentrated at 0 (see Theorem A.2), we don't need to worry if neurons move into this region. Neurons in $B_1 \cup B_{2,1} \cup B_{2,2} \cup B_{2,3}$ might exit "bad type" regime and become "harmless" or "good" (if the neuron becomes order-aligned), which will do no harm to the performance of the model. If they stay in $B_1 \cup B_{2,1} \cup B_{2,2} \cup B_{2,3}$, the following theorem shows that the separation $m_0^i - m_1^i$ can be upper bounded by initialization. In fact, Theorem A.4 shows that $m_0^i - m_1^i$ is proportional to $|\alpha_i - \alpha_i'||\beta_i - \beta_i'|$. The next theorem shows that both $|\alpha_i - \alpha_i'|$ and $|\beta_i - \beta_i'|$ shrink throughout coordinate descent. The worst situation is that the magnitude of $|\alpha_i - \alpha_i'|$ and $|\beta_i - \beta_i'|$ of neurons in $B_3$ increase and move into $B_1$ or $B_2$ at certain epoch. From Theorem B.7 we see that the magnitude can only increase by a limited rate (we can see this more explicitly in Theorem 6.2).

**Theorem B.9.** *If* $(\alpha_i^{(k)}, \alpha_i'^{(k)}, \beta_i^{(k)}, \beta_i'^{(k)})$ *is in* $B_1 \cup B_{2,1} \cup B_{2,2} \cup B_{2,3}$ *then with probability* $\geq 1 - O(\frac{1}{n^2})$ *over the SBM randomness,* $|\alpha_i^{(k+1)} - \alpha_i'^{(k+1)}| \leq |\alpha_i^{(k)} - \alpha_i'^{(k)}|$, $|\beta_i^{(k+1)} - \beta_i'^{(k+1)}| \leq |\beta_i^{(k)} - \beta_i'^{(k)}|$.

*Proof.* In $B_1$ and $B_{2,1}$, by (12) and (13), $\frac{\partial L}{\partial \alpha_i^{(k)}} > 0 > \frac{\partial L}{\partial \alpha_i'^{(k)}}$, then $\alpha_i^{(k+1)} < \alpha_i^{(k)}, \alpha_i'^{(k+1)} > \alpha_i'^{(k)}$, so $|\alpha_i^{(k+1)} - \alpha_i'^{(k+1)}| \leq |\alpha_i^{(k)} - \alpha_i'^{(k)}|$. Similarly, by (14), $\frac{\partial L}{\partial \beta_i^{(k)}} = -\frac{\partial L}{\partial \beta_i'^{(k)}} < 0$, so $|\beta_i^{(k+1)} - \beta_i'^{(k+1)}| \leq |\beta_i^{(k)} - \beta_i'^{(k)}|$ (Note that $\alpha_i^{(k)} > \alpha_i'^{(k)}, \beta_i^{(k)} < \beta_i'^{(k)}$).

In $B_{2,2}$, from (15) and (16), we have $\frac{\partial L}{\partial \alpha_i^{(k)}} > \frac{\partial L}{\partial \alpha_i'^{(k)}} > 0$, so $|\alpha_i^{(k+1)} - \alpha_i'^{(k+1)}| \leq |\alpha_i^{(k)} - \alpha_i'^{(k)}|$. On the other hand, $\frac{\partial L}{\partial \beta_i^{(k)}} < 0 < \frac{\partial L}{\partial \beta_i'^{(k)}}$, so $\beta_i^{(k+1)} > \beta_i^{(k)}, \beta_i'^{(k+1)} < \beta_i'^{(k)}$ and $|\beta_i^{(k+1)} - \beta_i'^{(k+1)}| \leq |\beta_i^{(k)} - \beta_i'^{(k)}|$. In $B_{2,3}$, by (24), $\frac{\partial L}{\partial \alpha_i^{(k)}} - \frac{\partial L}{\partial \alpha_i'^{(k)}} > 0$, so $|\alpha_i^{(k+1)} - \alpha_i'^{(k+1)}| \leq |\alpha_i^{(k)} - \alpha_i'^{(k)}|$. By (18), $\frac{\partial L}{\partial \beta_i^{(k)}} < 0 < \frac{\partial L}{\partial \beta_i'^{(k)}}$, so $|\beta_i^{(k+1)} - \beta_i'^{(k+1)}| \leq |\beta_i^{(k)} - \beta_i'^{(k)}|$. $\square$

### B.3 "Harmless Type" Neurons

Section 4 shows that there are two cases of "harmless type": $H_1$ and $H_2$. For neurons in $H_1$, the derivatives of parameters are estimated in (15), (16) and (17) (same as in $G_{2,2}$). We can have similar analysis as in $G_{2,2}$ and show that the inequality $\alpha_i > 0 > \alpha_i', \beta_i > \beta_i'$ can be preserved. Moreover $\left|\frac{\alpha_i}{\alpha_i'}\right|$ increases. So the neurons either stay in $H_1$ or become "good type" if $\left|\frac{\alpha_i}{\alpha_i'}\right| > 1$. In particular, neurons in $H_1$ do no harm to the performance of the model.

For neurons in $H_2$, $\mathbb{1}[\alpha_i t_0^y + \alpha_i' t_1^y \geq 0] = 0$, so the derivatives are all equal to 0. Therefore they are never updated. Meanwhile they don't affect the performance of the model since $\phi(\alpha_i t_0^y + \alpha_i' t_1^y) = 0$ and $\Delta_i = 0$.

## C Learning Guarantee

In this section, we prove Theorem 6.1, 6.2 and Lemma 6.3.

*Proof of Theorem 6.1.* We prove by contradiction. Suppose

$$\mathbb{P}[\Delta < 0|\ell(x) = 0] \geq 4\epsilon, \tag{27}$$

then

$$\begin{aligned}
\mathbb{E}[Z|\ell(x) = 0] &= \mathbb{E}[Z|\ell(x) = 0, \Delta < 0]\mathbb{P}[\Delta < 0|\ell(x) = 0] \\
&+ \mathbb{E}[Z|\ell(x) = 0, \Delta \geq 0]\mathbb{P}[\Delta \geq 0|\ell(x) = 0] \\
&\geq \frac{1}{2} \cdot 4\epsilon = 2\epsilon.
\end{aligned} \tag{28}$$

Furthermore, we claim that $\mu_0 < \delta$. In fact, if $\mu_0 \geq \delta$, since $\mathbb{P}[|\Delta - \mu_0| \leq \delta|\ell(x) = 0] \geq 1 - \epsilon$ by Corollary 4.2, and $\Delta \geq \mu_0 - \delta \geq 0$, we have

$$\mathbb{P}[\Delta \geq 0|\ell(x) = 0] \geq \mathbb{P}[|\Delta - \mu_0| \leq \delta|\ell(x) = 0] \geq 1 - \epsilon,$$

i.e. $\mathbb{P}[\Delta < 0|\ell(x) = 0] \leq \epsilon$, which contradicts (27).

Let $c := \mu_0 - \mu_1$, then $\mu_1 = \mu_0 - c < \delta - c$. Again, by Corollary 4.2, for $\ell(x) = 1, \Delta < \mu_1 + \delta$ with probability $\geq 1 - \epsilon$, we have $Z = \sigma(\Delta) < \sigma(\mu_1 + \delta) < \sigma(-c + 2\delta)$. Then

$$\begin{aligned}
\mathbb{E}[Z|\ell(x) = 1] &= \mathbb{E}[Z|\ell(x) = 1, |\Delta - \mu_1| < \delta]\mathbb{P}[|\Delta - \mu_1| < \delta|\ell(x) = 1] \\
&+ \mathbb{E}[Z|\ell(x) = 1, |\Delta - \mu_1| \geq \delta]\mathbb{P}[|\Delta - \mu_1| \geq \delta|\ell(x) = 1] \\
&< \sigma(-c + 2\delta) \cdot 1 + 1 \cdot \epsilon \\
&< \sigma\left(-\frac{c}{2}\right) + \epsilon.
\end{aligned}$$

The last step is due to $\delta = o(c)$. Since $\sigma(-\frac{c}{2}) < \frac{\epsilon}{2}$, $\mathbb{E}[Z|\ell(x) = 1] < \frac{3\epsilon}{2}$. Combine with (28),

$$\left| \mathbb{E}[Z|\ell(x) = 0] - \mathbb{E}[Z|\ell(x) = 1] \right| > \frac{\epsilon}{2}.$$

On the other hand, $\left| \mathbb{E}[\frac{\partial L}{\partial b_0}] \right| < \frac{\epsilon}{4}$ implies

$$\left| \mathbb{E}[Z|\ell(x) = 0] - \mathbb{E}[Z|\ell(x) = 1] \right| < \frac{\epsilon}{2},$$

which is a contradiction. So $\mathbb{P}[\Delta < 0|\ell(x) = 0] < 4\epsilon$. Similarly, $\mathbb{P}[\Delta > 0|\ell(x) = 1] < 4\epsilon$. $\qquad\square$

*Proof of Theorem 6.2.* If the $i$-th neuron is of "good type", from Corollary A.4, we find a uniform lower bound of $m_0^i - m_1^i$ in "good type" regimes. We have $\min\{\frac{p-q}{2}, \Lambda_3\} = \frac{p-q}{2}$. Next we estimate $\alpha_i - \alpha_i'$ and $\beta_i - \beta_i'$. Let $A_i^{(k)} := \alpha_i^{(k)} - \alpha_i'^{(k)}$, $B_i^{(k)} := \beta_i^{(k)} - \beta_i'^{(k)}$. We have

$$A_i^{(k)} = \alpha_i^{(k)} - \alpha_i'^{(k)} = \alpha_i^{(k-1)} - \alpha_i'^{(k-1)} - \eta_k \left( \frac{\partial L}{\partial \alpha_i^{(k-1)}} - \frac{\partial L}{\partial \alpha_i'^{(k-1)}} \right)$$

$$B_i^{(k)} = \beta_i^{(k)} - \beta_i'^{(k)} = \beta_i^{(k-1)} - \beta_i'^{(k-1)} - \eta_k \left( \frac{\partial L}{\partial \beta_i^{(k-1)}} - \frac{\partial L}{\partial \beta_i'^{(k-1)}} \right),$$

By Theorem B.7, in $G_1$ and $G_{2,1}$, with probability $\geq 1 - O(\frac{1}{n})$,

$$\frac{\partial L}{\partial \alpha_i} - \frac{\partial L}{\partial \alpha_i'} \leq -(\beta_i - \beta_i')\mathbb{E}[Z]\left( \left(\frac{p-q}{p+q}\right)^2 \lambda - O(\log^{-\frac{1}{2}} n)) \right) \leq -(\beta_i - \beta_i')\mathbb{E}[Z]\frac{\lambda}{2}\left(\frac{p-q}{p+q}\right)^2$$

$$\frac{\partial L}{\partial \beta_i} - \frac{\partial L}{\partial \beta_i'} \leq -(\alpha_i - \alpha_i')\mathbb{E}[Z]\left( \left(\frac{p-q}{p+q}\right)^2 \lambda - O(\log^{-\frac{1}{2}} n) \right) \leq -(\alpha_i - \alpha_i')\mathbb{E}[Z]\frac{\lambda}{2}\left(\frac{p-q}{p+q}\right)^2$$

so

$$A_i^{(k)} = A_i^{(k-1)} - \eta_k \left( \frac{\partial L}{\partial \alpha_i^{(k-1)}} - \frac{\partial L}{\partial \alpha_i'^{(k-1)}} \right) \geq A_i^{(k-1)} + \eta_k \mathbb{E}[Z^{(k)}]\frac{\lambda}{2}\left(\frac{p-q}{p+q}\right)^2 B_i^{(k-1)}$$

$$= A_i^{(k-1)} + \frac{\lambda}{2}\left(\frac{p-q}{p+q}\right)^2 B_i^{(k-1)}$$

$$B_i^{(k)} = B_i^{(k-1)} - \eta_k \left( \frac{\partial L}{\partial \beta_i^{(k-1)}} - \frac{\partial L}{\partial \beta_i'^{(k-1)}} \right) \geq B_i^{(k-1)} + \eta_k \mathbb{E}[Z^{(k)}]\frac{\lambda}{2}\left(\frac{p-q}{p+q}\right)^2 A_i^{(k-1)}$$

$$= B_i^{(k-1)} + \frac{\lambda}{2}\left(\frac{p-q}{p+q}\right)^2 A_i^{(k-1)}.$$

In matrix form:

$$\begin{pmatrix} A_i^{(k)} \\ B_i^{(k)} \end{pmatrix} \succeq \begin{pmatrix} 1 & \frac{\lambda}{2}\left(\frac{p-q}{p+q}\right)^2 \\ \frac{\lambda}{2}\left(\frac{p-q}{p+q}\right)^2 & 1 \end{pmatrix} \begin{pmatrix} A_i^{(k-1)} \\ B_i^{(k-1)} \end{pmatrix} \tag{29}$$

Similarly, in $G_{2,2}$:

$$\begin{pmatrix} A_i^{(k)} \\ B_i^{(k)} \end{pmatrix} \succeq \begin{pmatrix} 1 & \frac{\lambda}{4}\left(\frac{p-q}{p+q}\right)^2 \\ \frac{\lambda}{8}\left(\frac{p-q}{p+q}\right)^2 & 1 \end{pmatrix} \begin{pmatrix} A_i^{(k-1)} \\ B_i^{(k-1)} \end{pmatrix} \tag{30}$$

in $G_{2,3}$:

$$\begin{pmatrix} A_i^{(k)} \\ B_i^{(k)} \end{pmatrix} \succeq \begin{pmatrix} 1 & \frac{\lambda}{4}\left(\frac{p-q}{p+q}\right)^2 \\ \frac{\lambda(\Lambda_3 - \Lambda_2)}{2(p-q)}\left(\frac{p-q}{p+q}\right)^2 & 1 \end{pmatrix} \begin{pmatrix} A_i^{(k-1)} \\ B_i^{(k-1)} \end{pmatrix} \tag{31}$$

where $\Lambda_2 = \frac{pq\log^{-\frac{1}{3}} n}{(1+\log^{-\frac{1}{3}} n)p+q}$, $\Lambda_3 = \frac{(1-\log^{-\frac{1}{3}} n)p^2-q^2}{(1-\log^{-\frac{1}{3}} n)p+q}$. A uniform relation among (29), (30) and (31) can be given by (30). By eigenvalue decomposition, we have

$$A_i^{(k)} B_i^{(k)} \geq \frac{1}{4}\left( 1 + \frac{\sqrt{2}\lambda}{8}\left(\frac{p-q}{p+q}\right)^2 \right)^{2k} \left( 2^{-\frac{1}{8}} A_i^{(0)} + 2^{\frac{1}{8}} B_i^{(0)} \right)^2$$

$$\geq A_i^{(0)} B_i^{(0)} \left( 1 + \frac{\sqrt{2}\lambda}{8}\left(\frac{p-q}{p+q}\right)^2 \right)^{2k}.$$

Therefore we have a uniform lower bound of $m_0^i - m_1^i$ at the $k$-th epoch in "good type" regime:

$$m_0^i - m_1^i \geq A_i^{(0)} B_i^{(0)} \frac{\lambda}{2} \left(\frac{p-q}{p+q}\right)^2 \left(1 + \frac{\sqrt{2}\lambda}{8}\left(\frac{p-q}{p+q}\right)^2\right)^{2k}.$$

Next we consider the "bad type" regime. By Corollary A.4, we have lower bound of $m_0^i - m_1^i$ in $B_1$, $B_2$ and $B_3$ respectively. By Theorem B.9, in $B_1$ and $B_2$, $|\alpha_i - \alpha_i'|$ and $|\beta_i - \beta_i'|$ shrink. Moreover, since $\Lambda_1 > \Lambda_3$[5], we have a uniform lower bound of $m_0^i - m_1^i$ in $B_1$ and $B_2$:

$$m_0^i - m_1^i \geq -\left|A_i^{(0)} B_i^{(0)}\right| \frac{\lambda(p-q)\Lambda_1}{(p+q)^2} \geq -\left|A_i^{(0)} B_i^{(0)}\right| \frac{\lambda p(p-q)}{(p+q)^2},$$

since $\Lambda_1 = \frac{(1+\log^{-\frac{1}{3}} n)p^2 - q^2}{(1+\log^{-\frac{1}{3}} n)p + q} \leq \frac{2p^2 - q^2}{2p+q} \leq p$.

Next we show that $|\alpha_i - \alpha_i'|$ and $|\beta_i - \beta_i'|$ can only increase by a limited rate in $B_3$. From item 4 of Theorem B.7, we have

$$\frac{\partial L}{\partial \alpha_i} - \frac{\partial L}{\partial \alpha_i'} \geq -|\beta_i - \beta_i'| O(\tilde{\epsilon})$$

$$\frac{\partial L}{\partial \beta_i} - \frac{\partial L}{\partial \beta_i'} = 2\frac{\partial L}{\partial \beta_i} \leq O(\tilde{\epsilon}).$$

Therefore (note that $\beta_i < \beta_i'$)

$$A_i^{(k)} \leq A_i^{(k-1)} + \eta_k |B_i^{(k-1)}| O(\tilde{\epsilon})$$

$$|B_i^{(k)}| \leq |B_i^{(k-1)}| + \eta_k O(\tilde{\epsilon}).$$

Since $\mathbb{E}[Z^{(k)}] \geq \Omega(\epsilon)$[6], and $\tilde{\epsilon} = o(\epsilon), \epsilon^2 = O(\frac{1}{n})$, so $\eta_k O(\tilde{\epsilon}) \leq \eta_k \mathbb{E}[Z^{(k)}]\frac{1}{n} = \frac{1}{n}$. Suppose $A_i^{(k)} \geq O(1)$, otherwise, $A_i^{(k)}$ and $|B_i^{(k)}|$ increase by an even smaller rate. So we have

$$\begin{pmatrix} A_i^{(k)} \\ |B_i^{(k)}| \end{pmatrix} \preceq \begin{pmatrix} 1 & \frac{1}{n} \\ \frac{1}{n} & 1 \end{pmatrix} \begin{pmatrix} A_i^{(k-1)} \\ |B_i^{(k-1)}| \end{pmatrix}.$$

By eigenvalue decomposition, we have

$$|A_i^{(k)} B_i^{(k)}| \leq \frac{1}{2}(1 + \frac{1}{n})^{2k}\left(|A_i^{(0)}| + |B_i^{(0)}|\right)^2 \leq k\left(\left(A_i^{(0)}\right)^2 + \left(B_i^{(0)}\right)^2\right) \quad (n \to \infty).$$

We obtained the result for "harmless type" neurons directly from Corollary 4.3. □

*Proof of Lemma 6.3.* Since all parameters are independent standard normal random variables, we have

$$\mathbb{E}[(\alpha - \alpha')^2] = \mathbb{E}[(\beta - \beta')^2] = 2,$$
$$\mathrm{Var}[(\alpha - \alpha')^2] = \mathrm{Var}[(\beta - \beta')^2] = 8.$$

By Chebyshev's inequality we have

$$\mathbb{P}[\sum_{i=1}^{h} (\alpha_i - \alpha_i')^2 + (\beta_i - \beta_i')^2 \leq 5h] \geq 1 - O\left(\frac{1}{h}\right).$$

---

[5] $\frac{xp^2 - q^2}{xp+q}$ is monotonically increasing.
[6] Otherwise, the model already achieves high accuracy, see the proof of Theorem 2.2

For neurons initialized as "good type", we have

$$\mathbb{E}[\alpha - \alpha' | \alpha > \alpha', \alpha + \alpha' > 0] = \frac{2}{\sqrt{\pi}}$$

$$\text{Var}[\alpha - \alpha' | \alpha > \alpha', \alpha + \alpha' > 0] = 2 - \frac{1}{4\pi}$$

$$\mathbb{E}[\beta - \beta' | \beta > \beta'] = \frac{1}{\sqrt{\pi}}$$

$$\text{Var}[\beta - \beta' | \beta > \beta'] = 2 - \frac{1}{\pi}.$$

Let $\rho$ denote the probability that a neuron is initialized as "good type". By $G_1$, $G_2$ and symmetry, $\rho = 2\mathbb{P}[\alpha > \alpha', \alpha + \alpha' > 0, \beta > \beta']$. Since

$$\mathbb{P}[\alpha > \alpha', \alpha + \alpha'] = \frac{1}{4}, \ \mathbb{P}[\beta > \beta'] = \frac{1}{2},$$

we have $\rho = \frac{1}{4}$. By Chernoff bound, $\mathbb{P}[h_g \geq \frac{\rho}{2}h] \geq 1 - \exp\left(-\frac{\rho^2}{4}h\right)$, so $\mathbb{P}[h_g \geq \frac{h}{8}] \geq 1 - \exp\left(-\frac{h}{64}\right)$.
Also by Chebyshev's inequality,

$$\mathbb{P}[\sum_{\substack{\text{the } i\text{-th neuron} \\ \text{initialized as} \\ \text{``good type''}}} |\alpha_i - \alpha_i'||\beta_i - \beta_i'| \geq h_g\left(\frac{1}{2\pi} - k\right)\big| h_g \geq \frac{h}{8}] \geq 1 - \frac{4 - \frac{1}{4\pi^2}}{h_g k^2}.$$

Set $k = \frac{1}{10\pi}$,

$$\mathbb{P}[\sum_{\substack{\text{the } i\text{-th neuron} \\ \text{initialized as} \\ \text{``good type''}}} |\alpha_i - \alpha_i'||\beta_i - \beta_i'| \geq \frac{h}{80}\big| h_g \geq \frac{h}{8}] \geq 1 - O\left(\frac{1}{h}\right).$$

So we have

$$\mathbb{P}[\sum_{\substack{\text{the } i\text{-th neuron} \\ \text{initialized as} \\ \text{``good type''}}} |\alpha_i - \alpha_i'||\beta_i - \beta_i'| \geq \frac{h}{80}]$$

$$\geq \mathbb{P}[\sum_{\substack{\text{the } i\text{-th neuron} \\ \text{initialized as} \\ \text{``good type''}}} |\alpha_i - \alpha_i'||\beta_i - \beta_i'| \geq \frac{h}{80}\big| h_g \geq \frac{h}{8}]\mathbb{P}[h_g \geq \frac{h}{8}]$$

$$\geq \left(1 - O\left(\frac{1}{h}\right)\right)\left(1 - \exp\left(-\frac{h}{64}\right)\right)$$

$$\geq 1 - O\left(\frac{1}{h}\right).$$

$\square$

## D EXPERIMENTS ON DYNAMICS OF HIDDEN NEURONS

This experiment verifies our argument in Sections B.1, B.2, B.3 and Theorem 6.2 about the dynamics of hidden neurons. We set $h = 5, \lambda = 0.3$ and train the model on graphs sampled from SBM with $n = 1000, a = 1.0, b = 0.7$. The plot of accuracy and its distribution can be seen in Section 7. Here we plot the dynamics of all the 5 hidden neurons in Figure 4, with each row corresponding to one hidden neuron. In each plot, $x$-axis represents epoch and $y$-axis represents the value of neurons. The first column depicts $\alpha_i$ and $\alpha_i'$, the second column $|\frac{\alpha_i}{\alpha_i'}|$, the third column $|\alpha_i - \alpha_i'|$, the fourth column $\beta_i, \beta_i'$ and the last column $|\beta_i - \beta_i'|$. As shown in the figure, the first, second and fourth neurons are of "good" type satisfying $(G_2)$. Throughout training these neurons are preserved as "good" type: they're order-aligned, $|\frac{\alpha_i}{\alpha_i'}|$ is lowered bounded by 1, and both $|\alpha_i - \alpha_i'|, |\beta_i - \beta_i'|$ keeps increasing. All of these verify our argument in B.1. The third neuron is "harmless" satisfying $(H_2)$. As shown

in B.3, this neuron isn't updated and doesn't make contribution to the output. The fifth neuron is of "bad" type satisfying ($B_2$). Although $|\alpha_i - \alpha_i'|$ and $|\beta_i - \beta_i'|$ increase, but by comparing with the first, second and fourth row ("good" neurons), they increase at a much smaller rate. This verifies our result in Theorem 6.2.

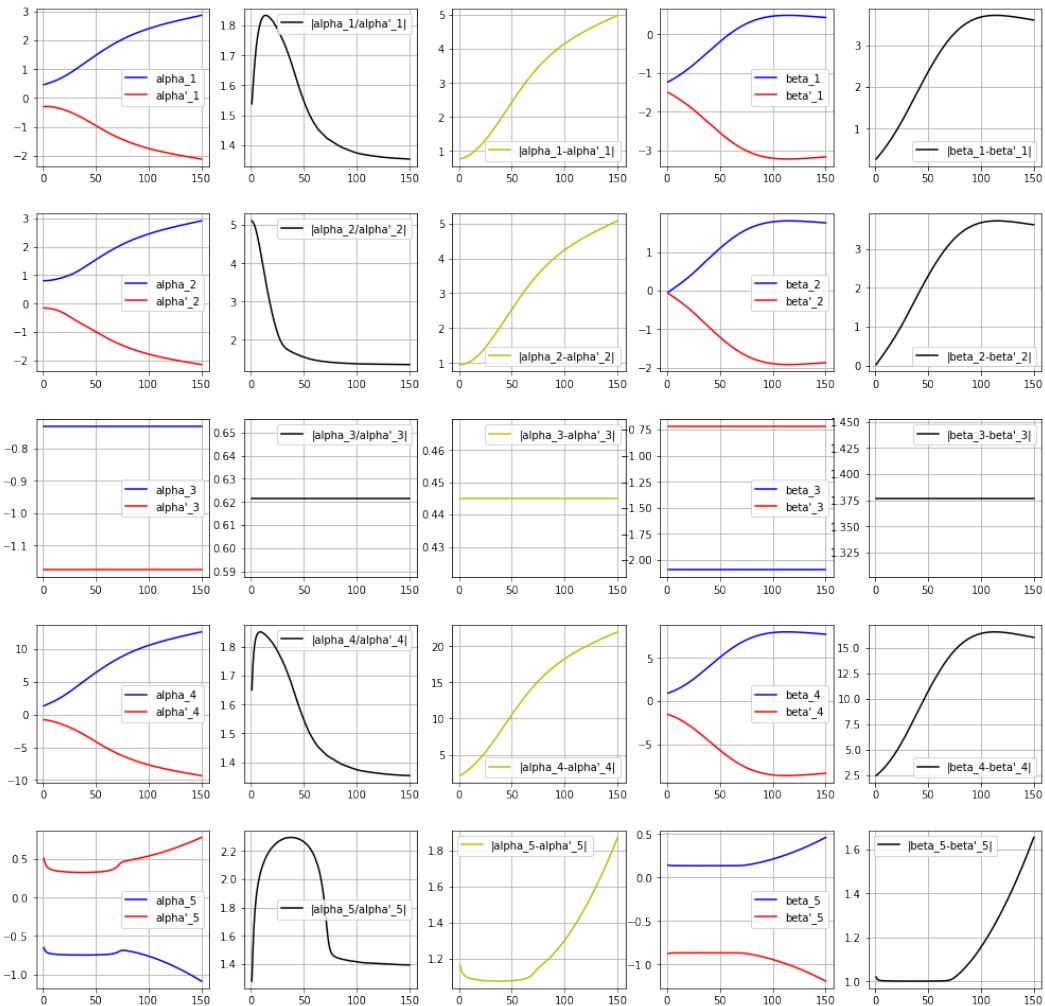

Figure 4: Dynamics of neurons of different types. The 1st, 2nd and 4th row correspond to neurons of "good" type satisfying ($G_2$). The 3rd row corresponds to "harmless" satisfying ($H_2$). The 5th row corresponds to "bad" type satisfying ($B_2$). Their dynamics verify our results in Sections B.1, B.2, B.3 and Theorem 6.2.

# E TABLE OF NOTATIONS

We list the notations used in this paper for readers' convenience.

| Notation | Definition |
|---|---|
| $n$ | number of vertices in a graph |
| $p$ | probability of intra-community connection |
| $q$ | probability of cross-community connection |
| $a$ | parameter for $p$ with $p = \frac{a \log^3 n}{n}$ |
| $b$ | parameter for $q$ with $q = \frac{b \log^3 n}{n}$ |
| $\lambda$ | probability of revealing the label of a vertex |
| $\ell(x)$ | label of vertex $x$ |
| $A$ | adjacency matrix of a graph |
| $\hat{A}$ | normalized adjacency matrix with self loop $\hat{A} = \frac{2}{n(p+q)}(A + I)$ |
| $X$ | input feature of a graph |
| $W^{(0)}$ | trainable weights in the first layer of GCN |
| $W^{(1)}$ | trainable weights in the second layer of GCN |
| $B$ | bias matrix of GCN with each row of $B$ being $[b_0, b_1]$ |
| $b_0$ | bias in the first component |
| $b_1$ | bias in the second component |
| $h$ | number of hidden features |
| $f_0$ | logit in the first component without bias |
| $f_1$ | logit in the second component without bias |
| $g_0$ | logit in the first component, $g_0 = f_0 + b_0$ |
| $g_1$ | logit in the second component, $g_1 = f_1 + b_1$ |
| $\Delta$ | difference between logit, $\Delta = g_0 - g_1 = f_0 - f_1 + b_0 - b_1$ |

