# OpenReview forum: "LEARNING GUARANTEES FOR GRAPH CONVOLUTIONAL NETWORKS ON THE STOCHASTIC BLOCK MODEL"
_ICLR.cc/2022/Conference — ICLR 2022 Poster_

### Official Review · Reviewer_15hz · 2021-10-29

**Correctness:** 3
**Technical Novelty And Significance:** 4
**Empirical Novelty And Significance:** 2
**Recommendation:** 6
**Confidence:** 3

**Main Review:**

The result is quite interesting and leveraging well-established graph models to provide guarantees for graph neural networks is certainly a promising direction.

I have the following comments for the authors:
1) The notation and presentation can be improved. For example:
- in page 3, the term $b$ is used first to determine the probabilities of the SBM and then to denote the bias.
- b_0 and b_1 are not introduced in page 3.
- The function $g$ is used in page 3 but introduced in page 4.
- A “bad type” neuron is mentioned in page 5 as satisfying B1 or B2 … what is B3 then?

2) Relation with classical detectability limits in SBMs.
The authors mention a few times that the problem of recovering the communities in an SBM has been well studied, but there are not direct connections between the regime of $p$ and $q$ studied here and what are the classical results for that case. Moreover, here the setting is different since in the classical setting one is given only a single SBM realization and here we are training with many realizations and then solving a semi-supervised problem. In principle, this means that a GCN could go beyond the classical detectability limits because the problem being solved is actually different. This is similar to the discussion in Section VII-C of “Exact Blind Community Detection from Signals on Multiple Graph” by Roddenberry et al. (although there are no GCNs involved there and indirect measurements of the topology are observed).

3) The main body of the paper is not self-contained.
To follow the logical flow of the paper one has to flip multiple times between appendix and the main text. The main text has corollaries of results in the appendix, there are several references to equations in the appendix, in page 6 there is even a reference to the fact that some of the results are stated in the "full version of the paper", and is unclear to me what the authors are referring to here.

4) In the experiments, the choices for $a$ and $b$ seem to be quite specific, is there any reason for these?
Also, in figure 3, from the description it looks like these histograms should sum up to 40, but they don't, what is going on here? E.g., for the center column, if we sum the are under the curve in the the top and middle plots these are very different, what is exactly being plot here then?

**Summary Of The Paper:**

This paper presents a theoretical result on performing label propagation in a stochastic block model (SBM) using a trained graph convolutional neural network (GCN). The idea is that by training a GCN to learn the community labels on several graphs drawn from an SBM, one can then claim some theoretical guarantees on the semi-supervised problem of determining the community labels of the nodes on a new graph (drawn from the same model) where only a subset of the nodes is labeled.


**Summary Of The Review:**

The main result of the paper is interesting, novel, and can motivate follow-up work. The presentation, on the other hand, should be improved.

---

> ### Author Response · Authors · 2021-11-19
> **feedback to the comment**
>
> We are very grateful to the reviewer for the helpful comments.
> 1. Questions about notations have been addressed on the top comment. On page 5, "bad type" neuron satisfies either $B_1, B_2$ or $B_3$, we are sorry for the typo.
> 2. Regarding the issue of detectability, our proof works in either training the model with one graph or on multiple graphs as we mentioned in the first paragraph on page 3, since we only made use of the distribution of vertices in a given SBM graph. When trained on one graph, classical results such as \url{https://arxiv.org/abs/1405.3267} shows that there is a sharp threshold phenomenon in the task of community recovery. The threshold $(\sqrt{p} - \sqrt{q})\sqrt{\frac{n}{\log n}} > \sqrt{2}$ shown in that paper actually clearly holds for our case (at sufficiently large values of $n$), since $p=\frac{a \log^3 n}{n}, q= \frac{b \log^3 n}{n}$ and $a > b$. But the goal of the paper is not to propose an algorithm to learn SBM models, we aim to prove the efficient learning of GCN on semi-supervised community detection tasks. It's an interesting problem to figure out the detectability limits when training on multiple graphs. We will leave it as our future work.
> 3. We have addressed the problem about reference on the top comment. The main text in the updated paper is more self contained. The "full version of the paper" on page 6 means "appendix". We are sorry for the confusion.
> 4. Questions about $a$ and $b$ in the experiments are also addressed on the top and new experimental results have been shown in the revision. The original Figure 3 is obtained from kde of the histogram. To avoid confusion, in the updated paper we replace it with the raw histogram where in each plot the value of the bars indeed sum up to 40. .

---

### Official Review · Reviewer_suTC · 2021-11-02

**Correctness:** 3
**Technical Novelty And Significance:** 4
**Empirical Novelty And Significance:** 2
**Recommendation:** 5
**Confidence:** 4

**Main Review:**

Strengths :-
1) The problem the authors address in this work is highly significant. They provide the first provable guarantees for GCNs for semi-supervised community detection tasks. This paper can accelerate much needed theoretical work in the graph neural networks.

Weaknesses :-
1) The presentation of the paper is bad and really hard to follow.
i) The same variable name is used in multiple avenues to mean different meanings which makes it hard to ascertain what it means when used i.e. b.
ii) The notation about stochastic block models (SBMs) is unclear. SBMs are specified via providing the partition of nodes which I could not find in the description.
iii) Some of the references were confusing.
The authors should have added a table for their notations which would have it much easier to follow. In case they were short on space. they could have added it in the appendix/supplementary materials. The current version of the draft needs some work to make it more digestible and easier to follow.

2) The experiments were unclear to me i.e. why did the authors choose the values of a and b and where did they get these values from. Will any values of a and b work as long as a > b i.e. Assumption 2.1 or are there only specific values we can use.
3) The experiments lack any discussion or analysis with regards to the results and how they are tied to the theory introduced in the paper.


**Summary Of The Paper:**

In this paper, the authors present the first provable guarantees for Graph Convolutional Networks (GCNs) for semi-supervised
community detection tasks. The authors demonstrate that with high probability over the initialization and training data used, a GCN will efficiently learn to detect communities on graphs drawn from a stochastic block model (SBM). Empirical results demonstrate the efficacy of their results.

**Summary Of The Review:**

In this work, the authors provide the first provable guarantees for the performance of GCNs for semi-supervised community detection tasks, which is highly significant. The presentation and notations used in the paper requires quite a bit of work to make them easier to follow for readers. The experiments were unclear to me as well as lack discussion or analysis. My recommendation was split between 5: marginally below the acceptance threshold and 6: marginally above the acceptance threshold and ultimately I decided to go with the former.

---

> ### Author Response · Authors · 2021-11-19
> **About presentation and experiments**
>
> We are very grateful to the reviewer for the helpful review.
> 1. Questions about presentation have been addressed in the top comment. i) We have replaced $b$ with $B$. ii) SBM is redefined in the updated paper. iii) References are adjusted to make the main text more self contained. A table of notations is added to the end of the appendix.
> 2. Questions about $a$ and $b$ are also addressed on the top. We have fixed $a=1.0, b=0.7$ to redo the experiments. In our proof and experiments, any values of $a$ and $b$ with $a>b$ would work. As shown in \url{https://arxiv.org/abs/1405.3267}, there is a sharp threshold phenomenon in the task of community recovery. The threshold $(\sqrt{p} - \sqrt{q})\sqrt{\frac{n}{\log n}} > \sqrt{2}$ shown in that paper actually clearly holds for our case (at sufficiently large values of $n$), since $p=\frac{a \log^3 n}{n}, q= \frac{b \log^3 n}{n}$ and $a > b$. As we mentioned in the paper, smaller $p$ and $q$ case is more interesting and will be left as forthcoming work.
> 3. The experiments are to verify the main result. The first experiment is to verify the relation between accuracy and $n$ (size of graph). The theorem shows that when $n > \Omega(\frac{1}{\epsilon})^2$, the model achieves $1-4\epsilon$ accuracy (when there are enough hidden neurons). Figure 2 shows that the accuracy increases uniformly when $n$ increases as expected from the theory. The second experiment shows the effect of $n$ and $h$ (number of hidden neurons). The theorem shows that when $h>\frac{1}{\delta}$, the model achieves high accuracy with probability $1-\delta$. This is what we can observe from Figure 3. For each row, as $h$ increases, we have higher probability to have high accuracy. On the other hand, for each column, when $n$ increases, the accuracy becomes higher.  In the updated paper, we further add an experiment to the appendix to verify our argument about the dynamics of hidden neurons. We are glad to add these discussion in our revision.

---

### Official Review · Reviewer_EjwU · 2021-11-05

**Correctness:** 3
**Technical Novelty And Significance:** 3
**Empirical Novelty And Significance:** 3
**Recommendation:** 8
**Confidence:** 3

**Main Review:**


The contribution is interesting although some of the aspects of the paper could be improved (see my comments below). Not all the sections are clear (e.g. the need for coordinate descent could be better explained). All in all, I think it can get published after some minor revisions.

**Summary Of The Paper:**

The authors provide guarantees (in term of the probability of correctly labelling any given vertex) regarding the learning of communities in the stochastic block model.  The proof technique relies on the analysis of three types of neurons : the so-called "good" neurons which contribute positively to the correct labelling, the so-called "harmless" neurons, whose effect on the classification can be neglected and the "bad" neurons which contribute negatively to the classification. The authors show that when initializing the paramters from a standard normal distribution, the number of neurons of the good type is sufficiently large to guarantee exact recovery of the missing label with high probabilty.

**Summary Of The Review:**

===============================
General comment
===============================

The transition between the general result of Theorem connecting the values of the neurons and the prediction accuracy on the one hand, and the result of Theorems 7.1 and 7.2 could be made more clear. It would be better to explain in plain english somewhere that you are interested in (1) deriving a bound on the prediction accuracy based on the value of the neurons and (2) show how that bound can be controled from the initialization. As an example, instead of section 4, which is useless and does not clearly introduce section 7, I would put a short paragraph, in the general introduction explaining the two aforementioned steps. In a few lines, I think that would clarify the whole paper.

===============================
Detailed comments
===============================

Abstract:
- I would be careful (remove or expand) with the sentence “This state of affairs is in contrast with the steady progress on the theoretical underpinnings of traditional dense and convolutional neural networks ”. It is not clear to me that a lot of progress has been made on the understanding of traditional convolutional neural networks

Introdution:

- Perhaps provide a word of explanation on the “Weisfeiler-Leman hierarchy”

Section 2:

- Your definition of the SBM is not very clear. I would say something like “In the setting of the SBM, the graph is built as follows. Two communities C_0 and C_1 are first created. , For any two vertices v_i and v_j, those vertices are then connected with probability q they belong to different communities and p if the belong to the same community”
- When you introduce b_0 and b_1 you should introduce b as well. In particular, the notations of section 3 (or at least some of those notations such as b) should already appear in section 2 when you introduce the model of the GNN.
- In section 2, page 3, when you discuss the balanced loss, the connection between the average of the derivative of the loss with respect to each of the b_0 and b_1 weights and the averages of Z over the vertices labeled as 0 and 1 is not clear.
- In your definition of Z, g0 and g1 are not defined. Are g_0 and g_1 defined as in section 3? then those definitions should appear earlier
- You initially mention parameters W^{(0)}, W^{(1)} and b as the trainable parameters but then only discuss coordinate descent for b0 and b1 which are not defined. This is not clear
- In Theorem 2.2, when you describe the accuracy, as correctly predicting the label of a single sample, I would recall the definition of lambda. In fact, lambda should be defined in the statement of Theorem 2.2
- The sentence “bigger λ generates bigger gap” is unclear. What do you mean? remove or expand
- generally speaking, the definition of g_0 and g_1 is not very clear. I would start by saying that g_j is defined from f_j and b_j. Then you can say g_0(x) denote the logit so that g_0(x) = log(p_0) + c  and  g_1(x) = log(p_1) +c  where p_0 and p_1 denote the probabilities of the vertex being classified in either of the two classes C_0 or C_1
- Don’t you make your life overly complicated by taking the softmax while you only have two communities ? Why not take the sigmoid with a single output p_0=p, assuming p_1 = 1-p ?

Section 4 :

- it does not really make sense to have a whole section to describe the paper structure. Put section 4 as the last paragraph of section 3, or add it as a subsection in 5

Section 5

- It would be good to have some additional intuition on conditions G_1 to H_2
- What if p=q in Corollary 5.3. Is that the reason why you take a>b in assumption 2.1. You should state that clearly. Why is that a problem ?
- The third remark is not very clear to me. I understand that we don’t want mu_0 and mu_1 to have the same sign as this would mean that the network is unable to discriminate between the classes but the output to the network is encoded in Delta not in mu so it is not clear to me that a small loss necessarily implies a small difference mu_0 - mu_1

Section 7

- In the statement of Theorem 7.1 I would explain the condition on mu_0 - mu_1 by adding a sentence of the form : “provided that the difference mu_0-mu_1 is large enough”, …
- The distinction between Theorem 7.2 and Corollary 5.3 is not clear. In particular, I want to make sure I understand the following: What do Theorem 5.1 and Corollary 5.3 assume in terms of the training ? Do they assume a 0 loss. This should be stated somewhere. The ambiguity also comes from the fact that you say that Theorem 7.2 refines Corollary 5.3. in what sense is Theorem 7.2 a refinement of Corollary 5.3? From what I understand Theorem 7.2 is just the application of corollary 5.3 to the dynamics of the neurons.
-  The sum in the second probability in the statement of lemma 7.3 is unclear. I guess you mean the sum over all neurons whose initialization satisfies G1 and G2 ?
- Proof of Theorem 2.2. Why can you assume that P(\ell(x) = 0) is 1/2 ? doesn’t that depend on the probabilities p and q ?

Section 8
- Figure 3 is not very intuitive. Why not add axes labels? Also why not indicate on the figure that n increases with the rows and h with the columns

A couple of additional typos and comments

- page 3, right above Theorem 2.2: “It is easy to see that Z > 1 if prediction is wrong “ —> “if the prediction is wrong”
- page 3, right above Theorem 2.2: “Then we update other parameters” —> “Then we update the other parameters”
- In the statement of corollary 5.3, you should replace the “i-the” neuron in each line with ”ith neuron”
- section 7: “In this section, we prove our main result that with high probability ” should be replaced with something like “In this section, we prove our main result which states that …” or “according to which ..”
- Section 8, first paragraph: “is able to recovery” —> “to recover”

---

> ### Author Response · Authors · 2021-11-19
> **feedback to reviews**
>
> We are grateful to the reviewer for the careful reading and resourceful comments.
>
> Section 1
> 1. A footnote is added to explain “Weisfeiler-Leman hierarchy”: Weisfeiler-Leman hierarchy is a polynomial-time iterative algorithm which provides a necessary but insufficient condition for graph isomorphism.
>
> Section 2
> 1. On page 15, we show that $\frac{\partial L}{\partial b_0} = -\frac{\partial L}{\partial b_1} = (-1)^{1-\ell(x)}Z$. Direct computation can give us the relation $\mathbb{E}[\frac{\partial L}{\partial b_0}] = -\mathbb{E}[\frac{\partial L}{\partial b_1}] = -\frac{1}{2}(\mathbb{E}[Z| \ell(x)=0] - \mathbb{E}[Z|\ell(x)=1])$, as shown on page 3.
> 2. The reason that "bigger $\lambda$ generates bigger gap" is due to equation (4) on page 8. As we can see, the separation between $\mu_0$ and $\mu_1$ is proportional to $\lambda$, so bigger $\lambda$ generates bigger gaps between $\mu_0$ and $\mu_1$. But since we treat $\lambda$ as a constant and omit it in the big-O notation in Theorem 2.2, we will remove that sentence to avoid confusion.
>
> Section 5
> 1. If $p=q$, the probability of inter-community and intra-community connection becomes the same and we cannot distinguish the two communities from the output value of the model.
> 2. The reviewer is correct that the prediction of the network is determined by $\Delta$. But $\Delta$ is highly concentrated at $\mu_0$ and $\mu_1$ respectively for $\ell(x)=0$ and 1 as shown in Corollary 5.2. To make correct prediction, we need $\mu_0 > 0$ for $\ell(x)=0$. Then $\Delta > 0$ with high probability, which implies that logit in the first component is bigger than in the second component, so we predict the label as 0. Similar interpretation holds for $\mu_1 < 0$. Furthermore, small loss implies big difference between $\mu_0$ and $\mu_1$.
> In fact, according to the cross-entropy loss shown on page 15, when $\ell(x)=0$, loss $=-\log \frac{\exp{(g_0)}}{\exp{(g_0)} +\exp{(g_1)}} = -\log \sigma(\Delta)$, where $\sigma$ is sigmoid function (note that $\Delta=g_0 - g_1$). So in order to have small loss, we want $\Delta$ to be as large as possible, i.e. we want $\mu_0$ to be large, since $\Delta$ is concentrated at $\mu_0$. Similarly, when $\ell(x)=1$, loss $= -\log \sigma(-\Delta)$. Small loss implies small $\mu_1$ (since $\Delta$ is concentrated at $\mu_1$ in this case). Therefore, we need $\mu_0$ and $\mu_1$ to be separated enough, this is what we show in Corollary 5.3 and Theorem 7.2: separation of output.
>
> Section 7
> 1. The aim of both Corollary 5.3 and Theorem 7.2 is to characterize the separation between $\mu_0$ and $\mu_1$. In Corollary 5.3, we don't assume anything about training. The result holds as long as the hidden neuron is of a particular type (this also applies to Theorem 5.1). The reviewer is correct that we apply dynamics of neurons to obtain Theorem 7.2 from Corollary 5.3. The reason we say Theorem 7.2 is a refinement of Corollary 5.3 is that the result in Theorem 7.2 only depends on the initialization of the hidden neurons, which gives us a more explicit scale of the separation between $\mu_0$ and $\mu_1$.
>
> 2. The reviewer's interpretation is correct about the summation in the second probability in Lemma 7.3.
>
> 3. Since we are considering a SBM graph with two equal-sized communities, a random chosen vertex has label 0 with $1/2$ probability.
>
> We thank the reviewer for other detailed comments about formulation of the paper, which we are happy to implement.

---

### Official Review · Reviewer_Pgr4 · 2021-11-08

**Correctness:** 2
**Technical Novelty And Significance:** 2
**Empirical Novelty And Significance:** 2
**Recommendation:** 5
**Confidence:** 3

**Main Review:**

The paper presents an interesting technical approach studying the concentration or separation of GCN outputs by understanding the dynamics of three parameter settings (good, bad, harmless). However, currently, the paper faces two major issues:

The main result is not informative enough and may contain erroneous claims. The probability $1 - O(1/h)$ is implicit and does not depend on $\epsilon$. It’s hardly (or impossible) to represent $h$ under both $n$ and $\epsilon$. It is because $n \geq A^2/\epsilon^2$ and $h \leq B n $ ($A, B are some constants$), we cannot say anything about the relation between $h$ and $\epsilon$. For example, given that $n$ is very big, $1/\epsilon^2$ is very small, we can pick $h$ any value that is either greater or smaller than $1/\epsilon^2$ but still smaller than $n$. Therefore, without establishing the result in PAC manners, the claim can not be convincing.

The experiments cannot justify the theoretical results. Clearly, the assumption states that $a, b$ are constant, $p, q$ are the function of $n$. It is expected to fix $a, b$ and consider multiple settings of $n$.  However, the experiment setup varies both $a$ and $b$ and seems to keep $p, q$ similar between configurations.

**Summary Of The Paper:**

The paper presents a theoretical analysis of two-layer graph convolutional networks (GCN). The main goal is to study the behaviors of GCNs when the inputs are random graphs generated by stochastic block models. The stochastic block models are constructed by three components: the number of vertices, intra-connection probability, and cross-connection probability. The paper assumes that intra-connection probability, and cross-connection probability should be functions of the number of vertices. The paper tries to establish the error bound w.r.t the number of vertices for the case that GCNs have the number of hidden features bounded by the number of vertices.

**Summary Of The Review:**

The main theorem seems interesting but can be problematic in the way it is interpreted. The experiments are designed in a way that does not support the theoretical claims. Since authors corrected the statement which fixes the condition of n and (especially) h, I raise the score to 5.

---

> ### Author Response · Authors · 2021-11-19
> **About PAC learning and experiments**
>
> 1. We are grateful for the reviewer for pointing out the issue of the statement of the main result. To make the result follow PAC manners, as we show in the comment on the top, we add a parameter $\delta$ to represent the probability parameter in the PAC statement. We reformulate our result as follows:
>
> For any $\epsilon>0$ and $\delta>0$, given a GCN model with $\frac{1}{\delta} \leq h \leq n$ hidden features and with parameters initialized independently from $N(0,1)$, if training graphs are sampled from SBM($n,p,q$) with $n \geq \max(\Omega(\frac{1}{\epsilon})^2, \Omega(\frac{1}{\delta}))$ and the label of each vertex revealed with probability $\lambda$, and if the model is trained by coordinate descent for $k = O(\log \log \frac{1}{\epsilon})$ epochs, then with probability $\geq 1- \delta$, the model achieves accuracy $\geq 1-4\epsilon$.
>
> The proof remains the same. In fact, with probability $\geq 1-O(\frac{1}{h})$, we have concentration of the output (Corollary 4.2 in the updated paper) and proper initialization (Lemma 6.3). So with probability $\geq 1-O(\frac{1}{h})$, the model achieves high accuracy. Since $h \geq \frac{1}{\delta}$, we have the probability $1-\delta$ as stated above. We have revised the statement of the main result and the remark following it on page 3.
>
> 2. We also thank the reviewer for comments on the experiments. We have addressed the question in the top comment and updated experimental results have been shown in the revision.

---

### Author Response · Authors · 2021-11-19
**Summary of revisions and common questions**

We are grateful to all the reviewers for the careful reading and insightful reviews. We have revised the paper accordingly and uploaded the new version. We summarize our revisions and address some common questions from reviewers here. We will answer some specific questions to each reviewer below.

1. We slightly modified the statement of the main theorem (Theorem 2.2) to make it follow PAC manners. We introduce a parameter $\delta$ to represent the probability of failed training. This probability can be controlled by the number of hidden neurons $h$. We also recall the definition of $\lambda$ in the statement. The proof of the theorem remains the same. The theorem is reformulated as follows:

Theorem: For any $\epsilon>0$ and $\delta>0$, given a GCN model with $\frac{1}{\delta} \leq h \leq n$ hidden features and with parameters initialized independently from $N(0,1)$, if training graphs are sampled from SBM($n,p,q$) with $n \geq \max(\Omega(\frac{1}{\epsilon})^2, \Omega(\frac{1}{\delta}))$ and the label of each vertex revealed with probability $\lambda$, and if the model is trained by coordinate descent for $k = O(\log \log \frac{1}{\epsilon})$ epochs, then with probability $\geq 1- \delta$, the model achieves accuracy $\geq 1-4\epsilon$.

2. We rewrite the definition of SBM on page 2 as follows to make it more clear.

"A graph is sampled from a SBM by first partitioning vertices into communities (with fixed or random sizes). Two vertices are connected with probability $p$ if they belong to the same community and probability $q$ if they belong to different communities. In this paper, we consider the case of an SBM with two equal-sized communities in which vertices have label 0 and 1 respectively. We denote the label of vertex $x$ by $\ell(x) \in \{0, 1\}$. The graphs are parameterized as SBM($n, p, q$) where $n$ is the number of vertices, $p$ is the probability of an intra-community connection, and $q$ is the probability of a cross-community connection."

3. We add an introduction to a classical detectability result on page 3 and shows its relation with our work. The detactability shows a sharp threshold phenomenon in the task of community recovery, i.e. $(\sqrt{p} - \sqrt{q})\sqrt{\frac{n}{\log n}} > \sqrt{2}.$ This threshold actually clearly holds for our case (at sufficiently large values of $n$), since $p=\frac{a \log^3 n}{n}, q= \frac{b \log^3 n}{n}$ and $a > b$. But, as we mentioned in the paper, the goal of this paper is not to propose an algorithm to learn SBM models, we aim to prove the efficient learning of GCN on semi-supervised community detection tasks.
4. The interpretation of bias term is moved from Section 2 to Section 3, since some used notations were introduced in Section 3. The notation of bias term is changed to $B$ to distinguish it from parameter $b$ in SBM.

5. Section 4 on the structure of the paper is incorporated into Section 3 and the title of Section 3 is changed from "Notation" to "Preliminaries".

6. References are adjusted in Sections 5 and 7. We mistakenly double labeled some theorems in both the main text and the appendix when we attempted to prove them. So some theorems were referred to the appendix when they should have been referred in the main text. We are sorry for the confusion and we have adjusted references to make the main text more self-contained.

7. The values of $a$ and $b$ in Section 8 are obtained from experiments we did before on a SBM($n, p, q$) with $p=1.5n^{-\frac{1}{7}}$ and $q=n^{-\frac{1}{7}}$. Now we have fixed $a, b$ and redone the experiments for varying $n$. Specifically, we set $a=1.0, b=0.7$ and run experiments for $n=250, 500, 1000$ respectively. Similar results as Figure 2 (acc. vs $n$) and Figure 3 (distribution of accuracy with respect to $h$ and $n$) on page 9 are obtained and we show them in the updated version of the paper. The original Figure 3 shows the kde (Kernel Density Estimate) of the histogram of accuracy and we have replaced it with the raw histogram to avoid confusion. More interpretation of the experimental results and their relation with the main theorem are added.

8. Proofs of Theorem 7.2 and Lemma 7.3 are added to the appendix which were missing in the original submission.

9. An experimental result about the dynamics of hidden neurons is added in the appendix to verify our argument in Section B in the appendix.

10. A table of notations is added to the end of the appendix for readers' convenience.

We thank again all the reviewers for the reviews to improve our paper.

---

### Decision · Program_Chairs · 2022-01-20

**Decision:**

Accept (Poster)

**Comment:**

This paper shows that (under some parameter range) graph convolutional networks learns communities in the stochastic block model. The result is clean, the proof techniques rely on partitioning neurons of three types and seems applicable to more general settings. The reviewers agree that the main theorems are interesting. There are some concerns among reviewers about the presentation of the paper, but many of them seem to be already addressed in the revised version, and I would recommend the authors to continue to improve the writing. There are also some concern about experiments, but the experiments are mostly used to validate the theorems, so clarifying how they are related would suffice. Overall the paper seems to have an interesting theoretical result on GCN.